# communications
# engineering

# GaNDLF: the generally nuanced deep learning framework for scalable end-to-end clinical workflows

Sarthak Pati [1,2,3,4,5], Siddhesh P. Thakur[2,3,4], İbrahim Ethem Hamamcı[2,6], Ujjwal Baid[2,3,4], Bhakti Baheti[2,3,4], Megh Bhalerao[2,4], Orhun Güley[5], Sofia Mouchtaris [2,7], David Lang[2,7,8], Spyridon Thermos[9], Karol Gotkowski[10], Camila González[10], Caleb Grenko[2,3,4], Alexander Getka [2,4], Brandon Edwards [11], Micah Sheller [1,11], Junwen Wu[11], Deepthi Karkada [11], Ravi Panchumarthy[11], Vinayak Ahluwalia[2,4], Chunrui Zou[2,4], Vishnu Bashyam[2,4], Yuemeng Li[2,4], Babak Haghighi[2,4], Rhea Chitalia [2,4], Shahira Abousamra[12], Tahsin M. Kurc [13], Aimilia Gastounioti [2,4,14], Sezgin Er [6], Mark Bergman[2,4], Joel H. Saltz [13], Yong Fan [2,4], Prashant Shah[11], Anirban Mukhopadhyay [10], Sotirios A. Tsaftaris[9], Bjoern Menze[5,15], Christos Davatzikos [2,4], Despina Kontos[2,4], Alexandros Karargyris[1,16], Renato Umeton [1,17,18,19,20], Peter Mattson[1,21] & Spyridon Bakas [1,2,3,4✉]

Deep Learning (DL) has the potential to optimize machine learning in both the scientific and clinical communities. However, greater expertise is required to develop DL algorithms, and the variability of implementations hinders their reproducibility, translation, and deployment. Here we present the community-driven Generally Nuanced Deep Learning Framework (GaNDLF), with the goal of lowering these barriers. GaNDLF makes the mechanism of DL development, training, and inference more stable, reproducible, interpretable, and scalable, without requiring an extensive technical background. GaNDLF aims to provide an end-to-end solution for all DL-related tasks in computational precision medicine. We demonstrate the ability of GaNDLF to analyze both radiology and histology images, with built-in support for *k*-fold cross-validation, data augmentation, multiple modalities and output classes. Our quantitative performance evaluation on numerous use cases, anatomies, and computational tasks supports GaNDLF as a robust application framework for deployment in clinical workflows.

[1] MLCommons, Medical Working Group, San Francisco, CA, USA. [2] Center For Artificial Intelligence And Data Science For Integrated Diagnostics (AI2D) and Center for Biomedical Image Computing and Analytics (CBICA), University of Pennsylvania, Philadelphia, PA, USA. [3] Department of Pathology and Laboratory Medicine, Perelman School of Medicine, University of Pennsylvania, Philadelphia, PA, USA. [4] Department of Radiology, Perelman School of Medicine, University of Pennsylvania, Philadelphia, PA, USA. [5] Department of Informatics, Technical University of Munich, Munich, Bavaria, Germany. [6] International School of Medicine, Istanbul Medipol University, Istanbul, Marmara, Turkey. [7] Department of Bioengineering, School of Engineering and Applied Science, University of Pennsylvania, Philadelphia, PA, USA. [8] Department of Mathematics, School of Arts and Sciences, University of Pennsylvania, Philadelphia, PA, USA. [9] Institute for Digital Communications, School of Engineering, The University of Edinburgh, Scotland, UK. [10] Department of Computer Science, Technical University of Darmstadt, Darmstadt, Hesse, Germany. [11] Intel Corporation, Santa Clara, CA, USA. [12] Department of Computer Science, Stony Brook University, Stony Brook, New York, NY, USA. [13] Department of Biomedical Informatics, Stony Brook University, Stony Brook, New York, NY, USA. [14] Mallinckrodt Institute of Radiology, Washington University School of Medicine, St. Louis, MO, USA. [15] Department of Quantitative Biomedicine, University of Zurich, Zurich, Switzerland. [16] Institute of Image-Guided Surgery of Strasbourg, Strasbourg, France. [17] Department of Informatics & Analytics, Dana-Farber Cancer Institute, Boston, MA, USA. [18] Department of Pathology and Laboratory Medicine, Weill Cornell Medicine, New York, NY, USA. [19] Department of Biostatistics, Harvard T.H. Chan School of Public Health, Boston, MA, USA. [20] Department of Biological Engineering, Department of Mechanical Engineering, Massachusetts Institute of Technology, Boston, MA, USA. [21] Google, Menlo Park, CA, USA. ✉email: sbakas@upenn.edu

Deep Learning (DL) describes a subset of Machine Learning (ML) algorithms built upon the concepts of neural networks[1]. Over the last decade, DL has shown great promise in various problem domains such as semantic segmentation[2–5], quantum physics[6], segmentation of regions of interest (such as tumors) in medical images[7–13], medical landmark detection[14,15], image registration[16,17], predictive modelling[18], among many others[19–21]. The majority of this vast research was enabled by the abundance of DL libraries made open-source and publicly available, with some of the major ones being TensorFlow (developed by Google) and PyTorch (by Facebook - originally developed as Caffe by the University of California at Berkeley), which represent the most widely used libraries facilitating DL research. Among the currently available libraries, PyTorch has demonstrated itself to be one of the most customizable and easily deployable through its robust and efficient C++ backend.

There have been various efforts by the medical imaging community towards addressing the clinical end-points of academic research, and packaging pre-coded/pre-trained models for data scientists to leverage and address clinical requirements (Fig. 1). However, all these efforts, resulting in numerous software packages, can confuse the less experienced user and result in endless hours of searching for the appropriate tool to use. To alleviate this situation, we hereby stratify these efforts into a set of well-defined categories to deepen the community's understanding (Fig. 2). Some of these efforts lie on one side of the spectrum and can be classified as "applications", since they focus on the end-

user, with powerful user interfaces (either graphical, or otherwise). Software packages on the other end of the spectrum can be stratified as "libraries", since they are built as a mechanism to access low-level machine functionality, while "toolkits" fall in between these two ends, and provide a layer of abstraction to enable research. Finally, "frameworks" fulfil various roles and attempt to provide a multitude of functions targeting both developers and end-users. Examples of such packages are the Medical Imaging Interaction Toolkit (MITK)[22] and the Cancer Imaging Phenomics Toolkit (CaPTk)[23]. GaNDLF is also a framework with a notably unique emphasis to DL. Figure 2 illustrates this stratification, while also providing some pertinent examples.

Some of these prior efforts are non-DL based, such as MITK[22], 3D Slicer[24], ITK-SNAP[25], and CaPTk[23]. While they have been lauded for their generalizability, they may fall short when it comes to competitive performance for specific tasks. Towards obtaining superior performance, various efforts concentrating on DL have been devised recently by the community, such as NiftyNet[26], DeepNeuro[27], ANTsPyNet[28], and DLTK[29], that are implemented in TensorFlow, as well as pymia[30], InnerEye[31], and MONAI[32], that are implemented in PyTorch. Additionally, there are specialized DL-based tools that cater to specific problems, such as segmentation[11,33–35], registration[36], or specific imaging domains, like PathML[37], TIAToolbox[38], HistomicsML[39], that focus on data engineering and enabling ML in computational pathology. However, all these applications and toolkits either (i) describe developer-focused tools targeting members of the

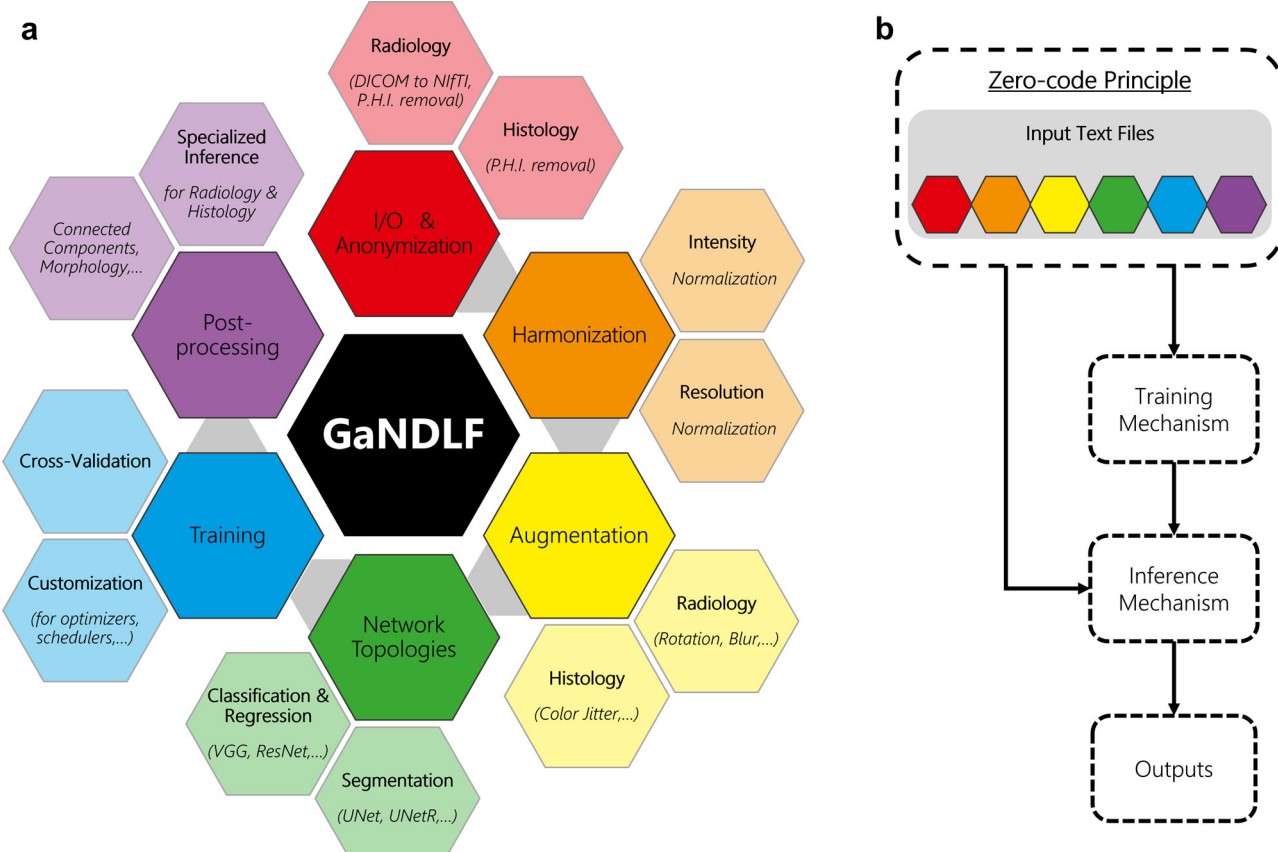

**Fig. 1 Current amalgamation of the functionality of GaNDLF. a** The entire functionality palette is focused to promote "zero/low-code" principles, and at the same time, each component in the major color groups (i.e., anonymization, harmonization, augmentation, network topologies, training, and post-processing) can be used independently to create customized solutions. The grey arrows represent the flow of operations for a user towards a "zero/low-code" principle for an entire computational training pipeline, starting with data I/O and ending with post-processing. **b** A high-level flowchart highlighting the "zero-code principle" entry point for the entire functionality palette of GaNDLF and their interactions throughout an AI clinical workflow, using the "zero/low-code" principle. A more comprehensive flowchart version is given in Fig. 4.

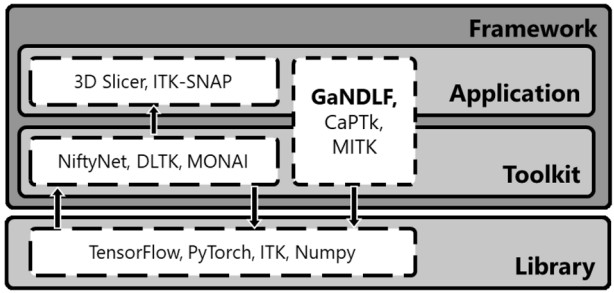

**Fig. 2 Schematic categorization of open-source software, with representative examples.** 'Libraries' focus on software developers offering access to low-level machine functionality. 'Toolkits' target computational experts and provide a layer of abstraction to enable research, by requiring users to write code to enable their functionality. 'Applications' focus on the non-computational end-user offering their functionality via user interfaces. 'Frameworks' fulfil both roles of 'Applications' and 'Toolkit', and provide a multitude of functions targeting both computational and non-computational end-users. Light gray represents software that a user interacts with on a lower level, and dark gray represents interaction using a command line or graphical interface.

advanced computational research community; (ii) can be difficult to grasp by researchers without sufficient experience in DL; (iii) do not make it easy for DL scientific developers to write their architectures in a generalizable way, allowing their application on problems spanning across domains; (iv) make it difficult to write reproducible training pipelines for different problem domains; (v) put the onus of training robust and generalizable models to the user's knowledge of the training mechanism and the dataset in question; (vi) lack a single end-to-end application programming interface (API) for training and inference that can span across various problem domains; or (vii) do not have appropriate level of interpretability or explainability functionality for researchers to garner meaningful insights into the training.

Here, we introduce the Generally Nuanced Deep Learning Framework (GaNDLF) as a community-driven open-source framework by MLCommons, which is an industry-academic partnership aiming to accelerate the adoption of machine learning innovation to benefit the larger community, to enable both clinical and computational researchers address various AI workloads (such as segmentation, regression, and classification), while producing robust AI models without requiring extensive computational experience. This is done by focusing on ensuring that AI algorithms and pipelines follow paradigms adhering to best practices established by the greater ML community, and leveraging existing collaborative efforts in the space (such as the MLCommons' MedPerf[40]). Such practices include: (i) nested cross-validation[41]; (ii) handling class imbalance[42]; and (iii) artificial augmentation of training data. Additionally, GaNDLF incorporates capabilities to handle end-to-end processing (i.e., pre- and post-processing steps) in a cohesive and reproducible manner to contribute towards democratizing AI in healthcare, while these best ML practices are at the forefront during training and inference. GaNDLF has been developed in PyTorch/Python as an abstraction layer that incorporates widely used open-source libraries and toolkits (such as MONAI[32]) that can help researchers generate robust AI models quickly and reliably, facilitating reproducibility and being consistent with the criteria of findability, accessibility, interoperability, and reusability (FAIR). Furthermore, the flexibility of its codebase permits GaNDLF to be used across modalities (e.g., 2D/3D radiology scans, and 2D multi-level histology whole slide images (WSI)), and has scope and functionality for integrating other clinical data (such as genomics and electronic health records) in the future,

thus taking current clinical diagnostics to the next frontier of quantitative integration.

## Results
To highlight the generalizability of the framework, GaNDLF was applied on both radiology and histology data for a variety of DL workloads/tasks (i.e., segmentation, regression, and classification) on multiple organ systems, imaging modalities, and various applications using numerous DL architectures. For each workload, we performed extensive performance evaluation using dedicated testing (or holdout[43]) datasets by averaging each model's training run in a cross-validated schema, ensuring stable model performance reporting without overfitting to a specific data split. Details regarding the experimental design of each application are shown in the Methods' Experimental Design section. The reported results for all the performed experiments are on the unseen testing (or holdout[43]) cohorts for each application, and collectively shown in Table 1.

**Segmentation workloads.** We applied GaNDLF to solve various segmentation problems on imaging acquired during standard clinical practice for multiple anatomical sites, comprising of brain, eyes, breast, lung, maxillofacial region, and colon. Numerous DL architectures, designed for segmentation workloads, were evaluated for multiple applications. These architectures include UNet, UNet with residual connections (ResUNet), Fully Convolutional network (FCN), and UNet with inception modules (see Methods section for details and Supplementary Figs. 1–10 for illustrations). Respective results are reported after quantitative performance evaluation based on Dice Similarity Coefficient ("Dice"). Note that GaNDLF offers the ability to generate other segmentation-specific metrics, such as the Hausdorff distance.

Several applications used brain Magnetic Resonance Imaging (MRI) scans, focusing on brain extraction (also known as skull-stripping)[11,44], boundary detection of histologically distinct brain tumor sub-regions[7–10], as well as comprehensive brain parcellation[45]. For brain extraction, we used each structural MRI volume as a separate independent input, with the goal of training a computational model that can segment the brain tissue region regardless of the input modality, and remove all non-brain tissue (e.g., neck, fat, eyeballs, and skull). In our analysis, we observed that the ResUNet architecture gave the best results, with average "Dice" of 0.98 ± 0.01. For brain tumor sub-regions, we considered the areas of necrosis, enhancing tumor, and peritumoral edematous/infiltrated tissue, following the convention of the International Brain Tumor Segmentation (BraTS) challenge[7–10]. To train these models, we used all four structural MRI volumes in tandem as input. For this application, the ResUNet architecture was again observed to give the best results with an average "Dice", across all the 3 sub-regions, of 0.71 ± 0.05. For brain parcellation, we segmented 133 fine-grained brain regions from the whole brain MRI scans[45]. In our analysis, we observed that ResUNet gave the most satisfactory results for the problem, with average "Dice" of 0.68 ± 0.15.

For the anatomical site of breast, we had two distinct applications. Firstly, we segmented the background, fatty breast tissue, and dense breast tissue from digital breast tomosynthesis scans[46]. Our experimentation resulted in the most optimal "Dice" scores using ResUNet, with an average of 0.94, 0.89, and 0.49, for each of the aforementioned regions, respectively, with an overall performance of 0.78 ± 0.09. Secondly, we segmented the structural tumor volume region from T1-weighted pre-contrast, peak-contrast and post-contrast injection scans using the ISPY-1 cohort[47]. We observed the best performance using ResUNet with an average "Dice" of 0.74 ± 0.01.

**Table 1 Results of various DL workloads using GaNDLF for multiple anatomies.**

| Task | Organ | Application | Dims | Input modalities (number): type | Output classes | Architecture | Metric | |
|---|---|---|---|---|---|---|---|---|
| | | | | | | | Type | Average value |
| Segmentation | Brain | Brain extraction | 3 | (1): T1, T1Gd, T2, T2-FLAIR as individual inputs | 1 | UNet | Dice | 0.97 ± 0.01 |
| | | | | | | ResUNet | Dice | 0.98 ± 0.01 |
| | | | | | | FCN | Dice | 0.97 ± 0.01 |
| | | Tumor sub-region segmentation | | (4): T1, T1Gd, T2, T2-FLAIR | 3 | UNet | Dice | 0.65 ± 0.05 |
| | | | | | | ResUNet | Dice | 0.71 ± 0.05 |
| | | | | | | FCN | Dice | 0.62 ± 0.05 |
| | | | | | | UInc | Dice | 0.64 ± 0.05 |
| | | Brain parcellation | | (1): T1 | 133 | ResUNet | Dice | 0.68 ± 0.15 |
| | | | | | | UNet | Dice | 0.57 ± 0.26 |
| | Breast | Breast segmentation | 3 | (1): Digital breast tomosynthesis | 3 | UNet | Dice | 0.78 ± 0.09 |
| | | Tumor segmentation | | (3): T1 pre, peak, and post-contrast injection | 1 | ResUNet | Dice | 0.74 ± 0.01 |
| | Lung | Lung field segmentation | 3 | (1): CT [Lung Cancer Screening] | 1 | ResUNet | Dice | 0.95 ± 0.02 |
| | | | | (1): CT [COVID-19] | | ResUNet | Dice | 0.97 ± 0.01 |
| | Eye | Fundus segmentation | 2 | (1): RGB Fundus Images | 1 | UNet | Dice | 0.85 ± 0.04 |
| | | | | | | ResUNet | Dice | 0.90 ± 0.05 |
| | | | | | | FCN | Dice | 0.81 ± 0.04 |
| | | | | | | UInc | Dice | 0.83 ± 0.03 |
| | Dental | Quadrant segmentation | 2 | (1): X-Ray | 4 | UNet | Dice | 0.91 ± 0.01 |
| | | | | | | ResUNet | Dice | 0.88 ± 0.01 |
| | | | | | | FCN | Dice | 0.85 ± 0.02 |
| | Colon | Colorectal cancer segmentation | 2 | (1): Histology H&E | 1 | ResUNet | Dice | 0.78 ± 0.03 |
| Regression | Brain | Age prediction | 2 | (1): T1 slices | 1 | Specialized VGG | MSE | 0.0141 ± 0.01 |
| Classification | Brain | EGFRvIII status prediction | 3 | (4): T1, T1Gd, T2, T2-FLAIR | 2 | VGG11 | Acc | 0.74 ± 0.08 |
| | Foot | Diabetic foot ulceration | 2 | (1): RGB Foot Images | 4 | VGG11 | Acc | 0.92 ± 0.01 |
| | | | | | | VGG16 | Acc | 0.90 ± 0.01 |
| | | | | | | VGG19 | Acc | 0.89 ± 0.01 |
| | | | | | | DenseNet121 | Acc | 0.87 ± 0.01 |
| | Pan-Cancer | TIL Prediction | 2 | (1): Histology H&E | 2 | ImageNet_ VGG16 | Acc | 0.89 ± 0.01 |

The "Task" showcases the workload type, "Organ" describes the organ system of the data, "Application" describes the use case for the trained model(s), "Dims" describe the dimensionality for each input modality, "Input Modalities" describes the total number of input modalities for the model to train on, "Output Classes" shows the number of classes the model should be predicting, "Architecture" describes the network topology, and "Metric" describes the type and average value of the selected metric on the testing/holdout dataset, and is "Dice" for segmentation tasks, Mean squared error or "MSE" for regression, and Balanced accuracy or "Acc" for classification.

For lung, we used low-dose Computed Tomography (CT) scans acquired for both lung cancer screening and COVID-19 assessment, with the intention to segment the lung field incorporating apparent healthy and abnormal tissue. Application of GaNDLF's ResUNet architecture on scans for both these applications (i.e., cancer screening and COVID-19 assessment), we observed "Dice" scores of 0.95 and 0.97, respectively.

For the anatomical site of the eyes, we segmented the fundus region in Red-Green-Blue (RGB) retinal scans[48], and observed that the ResUNet architecture gave the best results, with the average "Dice" coming to 0.71 ± 0.05.

For the anatomical region of maxillofacial, we have used panoramic dental x-ray images, with the goal of distinguishing and accurately segmenting the four quadrants considered in dental practice[49]. After training various DL architectures, we observed that UNet yielded the best results, with the average Dice coming to 0.91 ± 0.01.

Last but not least, and to evaluate GaNDLF's performance beyond radiology scans, we utilized histology digitized tissue sections (e.g., whole slide images (WSI)), stained for Hematoxylin and Eosin (H&E), of colorectal cancer by leveraging the publicly available dataset of the DigestPath challenge[50], with the intention of delineating the cancerous regions. Our results yield and average "Dice" of 0.78 ± 0.03 using ResUNet for a pre-defined testing data split.

**Regression**. For the DL workload of regression, we have used GaNDLF to solve a specifically targeted regression problem in brain MRI scans, focusing on predicting a surrogate index for brain age[51]. By virtue of the inherent flexibility in GaNDLF's design, we modified the VGG16 architecture to predict the age of a brain from a single MRI slice, and replicated previously reported results[51]. The input for this use case was based on 2D MRI slices of T1-weighted scans, and the output was the brain age. With an average mean squared error ("MSE") of 0.0141, the prediction quality of the models trained by GaNDLF was in line with the original publication[51], showcasing the flexibility of GaNDLF to successfully adapt to various problem domains.

**Classification**. We have further used GaNDLF to solve multiple classification problems, spanning different domains (e.g., radiology and histology), as well as various organ systems, including feet, brains, and pan-cancer histology images.

Specifically, for the anatomical location of brain, we have applied GaNDLF on 3D MRIs of patients diagnosed with *de novo* glioblastoma, to predict the EGFRvIII mutation status. The inputs for model training were structural MRI scans in tandem (passing all the scans together at once) to a VGG11 customized to perform computations in 3D directly, resulting in a best accuracy of $0.74 \pm 0.08$.

Furthering the application towards 2D RGB data, we have predicted different ulceration status of diabetic foot images from the DFU challenge[52] by passing each image as an input along with its ground truth label. We observed the best performance on VGG11[53] (that was randomly initialized instead of being pretrained on ImageNet), with a macro-F1 score of 0.561. Notably, the defined approach[53] was among the top-performing ones (ranked 5th) in the International in the DFU Challenge 2021 leaderboard (dfu-2021. grand-challenge.org/evaluation/challenge/leaderboard).

Finally, we used a dataset of histology digitized tissue sections stained for H&E, spanning across 12 anatomical sites. The problem at hand was to predict patches containing tumor-infiltrating lymphocytes (TIL)[54]. We observed the best-balanced classification accuracy of 0.89 using a VGG16 that was pre-trained on ImageNet[55] and customized for the specific problem.

## Discussion

We have introduced the Generally Nuanced Deep Learning Framework (GaNDLF), as an end-to-end solution for scalable clinical workflows, currently focused on (bio)medical imaging. GaNDLF provides a "zero/low-code" solution enabling both computational and non-computational experts to train robust DL models to tackle a variety of workloads/tasks in both 2D and 3D radiology and histology data, without worrying about details such as appropriate data splitting for training, validation, and testing, tackling class imbalances, and implementing various training strategies (e.g., loss functions, optimizers). Specifically, GaNDLF's contribution spans across its ability to: (i) process images of various domains, including both radiology scans and digitized histology WSIs; (ii) enable work on various workloads (i.e., segmentation, regression, and classification); (iii) offer built-in general-purpose functionality for augmentations and cross-validation; (iv) be evaluated on a multitude of applications; (v) enable parallel training by using generic high-performance computing protocols; (vi) integrate tools to promote the interpretability and explainability of DL networks, via M3D-CAM[56].

Our overarching goal is to enable clinical translation and applicability of AI, since specialized hardware (e.g., DL accelerator cards) is usually not considered for purchase by clinical entities in higher income countries, and altogether out of reach for clinics in lower-income countries. Towards this end, we have developed built-in model optimization support in GaNDLF to automatically generate optimized models after the training process is complete, allowing inference of these models on machines without requiring any specialized hardware, or large amounts of memory. We further envision the "model library" in GaNDLF to potentially be a phenomenal resource for pre-trained models and corresponding configurations to replicate training parameters for the scientific community in general. By ensuring that the model library contains information beyond just the trained model weights, but also additional metadata, trained models through GaNDLF will remain reproducible through code changes. GaNDLF is a fully self-contained DL framework that has various abstraction layers to enable researchers to produce and contribute robust DL models with absolutely zero knowledge of DL or coding experience.

The concepts of "zero-" and "low-" code principles in software development have recently been introduced, targeting different user groups. In essence, the "zero-code" principle revolves around allowing users to build solutions without writing any code, whereas the "low-code" principle allows customization of the provided solution with minimal programming. GaNDLF follows these zero/low-code principles and enables targeting a dual audience type: (i) non-computational experts, by providing building blocks for conducting DL analyses by leveraging their domain expertise without the need for any programming skills; (ii) DL researchers, allowing for harmonized I/O (i.e., common data loaders enabling the main focus be kept on the algorithmic development), as well as leveraging or extending existing capabilities to create custom solutions. For a non-computational researcher, GaNDLF ensures the easy creation of robust models using various DL architectures, and built-in automatically triggered ML principles, that can be used for scientific research and method discovery, including the potential for aggregating results from various models, which has been shown to provide greater accuracy[7,10]. For DL researchers/developers, GaNDLF provides a mechanism for creating customized solutions, robust evaluation of their methods across a wide array of medical datasets that span across dimensions, channels/modalities, and prediction classes, as well as to conduct a comparative quantitative performance evaluation of their algorithm against well-established built-in network architectures, including, but not limited to, UNet[57], UNetR (UNet with transformer encoding)[58], ResNet[59], and EfficientNet[60]. Furthermore, GaNDLF provides the means to DL researchers/developers to distribute their methods in a reproducible way to the wider community, thereby expanding their application across various problem domains with relative ease, and providing re-usable components (Fig. 1) that can be combined to create customized solutions. Ideally, we anticipate the best results when both these groups of the scientific and clinical community bring their expertise together to further our understanding of healthcare. Towards this end, GaNDLF can provide a common frame of reference for both these user groups. By creating tools standardized within the same infrastructure (GaNDLF) for the entire community to leverage, we anticipate the cost and time of creating algorithms to be substantially reduced and hence put efforts in meaningfully translating methods into the clinical practice rather than trying to identify and/or make a tool to work.

The modularity of the software stack is highlighted by large-scale studies of specific focus on federated learning (FL) that GaNDLF has facilitated, beyond the results shown in this manuscript. The FL-specific functionality is provided by its integration to work in conjunction with the Open Federated Learning (OpenFL) library[61]. Further integration with other community-driven efforts, such as MedPerf[40] (medperf.org) of MLCommons (mlcommons.org), would increase the applicability of GaNDLF towards federated learning applications[35,62]. GaNDLF has notably been used to orchestrate the Federated Tumor Segmentation (FeTS) Challenge[62], which represents the first-ever computational challenge on FL, targeting (i) the development of novel aggregation methods for federated training, and (ii) the federated evaluation of algorithms "in-the-wild", to assess algorithmic robustness to distribution shifts between medical institutions. Moreover, GaNDLF's codebase has facilitated components of the largest to-date real-world FL study (i.e., the FeTS Initiative[35] - www.fets.ai), involving data from 71 geographically-distinct collaborating sites to develop a DL model to detect boundaries of intrinsic sub-regions for the rare disease of glioblastoma in mpMRI scans. Finally, indicating its joint

ability with OpenFL to address workloads in various domains, the GaNDLF-OpenFL integration has enabled an FL histology study on identifying TILs in WSIs from numerous anatomical sites[63].

One of GaNDLF's core tenets is to enable work across domains, currently spanning radiology (e.g., MRI, CT) and histology (e.g., H&E-stained slides), including specialized pre-processing functionalities for each. The notable difference between these images is the relatively small resolution and size of radiology scans (typically occupying a few megabytes of disk space), compared with the histology WSI that are described by relatively large resolution (150 K × 150 K pixels) and size, where a single WSI can occupy 40–50 gigabytes. GaNDLF enables researchers to use a single framework across virtually all medical imaging modalities without performing any additional coding, thereby enabling future studies that rely on integrative diagnostics. Owing to the flexibility of the data loading mechanism in GaNDLF, it could also be possible to integrate other data types (such as genomic or healthcare records) into a model towards further contributing in the field of personalized medicine.

Although GaNDLF has been evaluated across imaging modalities using single inputs (i.e., either a single radiology or histology image) or with multi-channel support (i.e., multiple MRI sequences considered in-tandem), so far, its application has been limited to workloads related to segmentation, regression, and classification, but not towards synthesis, semi/self-supervised training, or physics-informed modeling. Expanding the application areas would further bolster the applicability of the framework. Additionally, application to datasets representing analysis of 4D images (such as dynamic sequences or multi-spectral imaging) has not yet been evaluated. Also, a mechanism to enable aggregation of various models (i.e., train/infer models of different architectures concurrently) is not present, which have generally shown to produce better results[7–10,33]. Mechanisms that enable AutoML[64] and other network architecture search (NAS) techniques[65] are tremendously powerful tools that create robust models, but are currently not supported in GaNDLF. Finally, application of GaNDLF to other data types, such as genomics or electronic health records (EHR), which would allow GaNDLF to further inform and aid clinical decision-making by training multi-modal models, has not been fully explored yet but it is considered as current work in progress.

To facilitate clinical applicability, reproducibility, and translation, in the domain of healthcare AI, published research is essential to adhere to well-accepted reporting criteria. Some of these criteria are: i) CLAIM (Checklist for Artificial Intelligence in Medical Imaging)[66], which outlines the information that authors of medical-imaging AI articles should provide, ii) STARD-AI, which is the AI-specific version of the Standards for Reporting of Diagnostic Accuracy Study (STARD) checklist[67], and aims to address challenges related to the original STARD checklist related to the utilization of AI models, iii) TRIPOD-AI and PROBAST-AI, which are the AI versions of the TRIPOD (Transparent Reporting of a multivariable prediction model of Individual Prognosis Or Diagnosis) statement and the PROBAST (Prediction model Risk Of Bias ASsessment Tool)[68], and aim to provide standards both for reporting but also for Risk of Bias assessment, raising awareness of the importance in meta-analyses dealing with AI studies, iv) CONSORT-AI and SPIRIT-AI, which are the AI extensions of the CONSORT (Consolidated Standards of Reporting Trials) and SPIRIT (Standard Protocol Items: Recommendations for Interventional Trials), providing guidance for reporting randomized clinical trials[69], v) MI-CLAIM (Minimum Information about Clinical Artificial Intelligence Modelling)[70], which focuses on the clinical impact and the technical reproducibility of clinically relevant AI studies, vi) MINIMAR (MINimum Information for Medical AI Reporting)[71], which sets the reporting

standards for medical AI applications by specifying the minimum information that AI manuscripts should include, and vii) Radiomics Quality Score (RQS)[72], which outlines 16 criteria by which to judge the quality of a publication on radiomics[73].

In conclusion, this manuscript describes GenerAlly Nuanced Deep Learning Framework ("GaNDLF"), a stand-alone package that provides end-to-end functionality facilitating transparent, robust, reproducible, and deployable DL research. Due to its flexible software architecture, it is possible to either leverage certain parts of GaNDLF in other applications/toolkits, or leverage functions of other toolkits (e.g., MONAI) and libraries to incorporate them within the holistic functionality of GaNDLF. Furthermore, GaNDLF could partner with container-based platforms beyond MedPerf[40] (such as the BraTS algorithmic repository, or ModelHub.AI) towards a structured dissemination of DL models to the research community. As all development is open-sourced in github.com/mlcommons/GaNDLF, with robust continuous integration and code vulnerability testing through Dependabot, contributions from the community will ensure that this framework continues building ties to other packages quickly and reliably for end users. Finally, by creating tools standardized within the same infrastructure (GaNDLF) for the entire community to leverage, we anticipate the cost and time of creating algorithms to be substantially reduced and hence put efforts in meaningfully translating methods into the clinical practice rather than trying to make a tool to work.

## Methods

**Pre-processing.** Providing robust pre-processing techniques that are widely applicable to (bio)medical data, is critical for such a general-purpose framework to succeed. GaNDLF offers most of the pre-processing techniques already reported in the literature, leveraging the capabilities of basic standardized pre-processing routines from ITK[74,75], and advanced pre-processing functionality from the CaPTk[23,76–79]. The main pre-processing steps for data curation (including harmonization and normalization) are described below.

1. Anonymization:

   - Radiology Images: Since the DICOM format[80] is the standard for radiology images, GaNDLF has provisions to remove all identifiable fields from the DICOM metadata, as well as a conversion to the Neuroimaging Informatics Technology Initiative (NIfTI) file format[81], which completely removes all extraneous metadata fields.
   - Histology Images: Most WSIs include metadata which could contain protected health information, and GaNDLF can remove such fields from the file header. This works for multiple formats defined by the Open Microscopy Environment standard[82], such as TIFF, SVS, and MRXS.

2. Data harmonization:

   - Voxel-resolution harmonization: To ensure that the physical definition of the input data is in a common space (for example, all images can have the voxel resolution of $I_{res} = [1.0, 1.0, 2.0]$).
   - Image-resolution harmonization: To ensure that the input data has the same image dimensions (for example, all images can be resampled to $I_{dim} = [240, 240, 155]$).

3. Intensity normalization:

   - Thresholding: To consider pixel/voxel values that belong to a specific intensity range and ignore values below/above this range, by making them equal to zero (Eq. (1)):

$$x_i = \begin{cases} 0 & x_i < threshold_{min} \\ 0 & x_i > threshold_{max} \\ x_i & otherwise \end{cases} \quad (1)$$

   - Clipping: To consider pixel/voxel values that belong to a specific intensity range and convert values below/above this range, by making them equal to the minimum/maximum threshold, respectively (Eq. (2)):

$$x_i = \begin{cases} threshold_{min} & x_i < threshold_{min} \\ threshold_{max} & x_i > threshold_{max} \\ x_i & otherwise \end{cases} \quad (2)$$

   - Rescaling: To consider all pixel/voxel values after converting them to a common profile (for example, all input images are rescaled to [0, 1]).

- Z-score normalization: A widely used technique for data normalization in medical imaging[83,84], that preserves the complete signal of the input image by subtracting the mean and then dividing by the standard deviation of the complete intensity range found in this image. Notably, the application of z-score normalization through GaNDLF can occur either on the full image or only within a masked region of interest, adding to the overall flexibility of this transform.
- Histogram Standardization[85] ensures harmonization of intensity profiles of input images based on a template (or reference) image. Different options are available to the user, such as histogram matching[85], adaptive histogram equalization[86] and global histogram equalization. Normalization methods specifically designed for WSIs that calculate stain vectors are also available, and these include methods from Vahadane[87], Ruifork[88], and Macenko[89].

**Data augmentation.** DL methods are well-known for being extremely data hungry[90,91] and in medical imaging, data is scarce because of various technical, privacy, cultural/ownership concerns, as well as data protection regulatory requirements, such as those set by the Health Insurance Portability and Accountability Act (HIPAA) of the United States[92] and the European General Data Protection Regulation (GDPR)[93]. This necessitates the addition of robust data augmentation techniques[94] into the training data, so that models can gain knowledge from larger datasets and hence be more generalizable to unseen data[95].

GaNDLF leverages existing robust data augmentation packages, such as TorchIO[96] and Albumentations[97], to provide augmentation transformations in a PyTorch-based mechanism. GaNDLF also stores image metadata (such as affine transform, origin, resolution), which is critical for maintaining correct physical coordinate definition of radiology scans. More details on the available types of augmentations through GaNDLF are shown in the Supplementary Notes 1: Details of Data Augmentation (Supplementary Table 1), and examples of their effects are illustrated in Fig. 3, using a brain tumor T2-FLAIR MRI scan from the BraTS challenge dataset[7–10].

**Training mechanism.** The overall pipeline of the training procedure offered in GaNDLF is illustrated in Fig. 4a, and focuses on stability and robustness for the user to generate reproducible results, and clinically-deployable models. Figure 4b showcases the overall software stack. The data flow of GaNDLF leverages 2 main ideas that allow efficient processing of large datasets (such as histology images or large 3D volumes): (i) patch-based training and inference, which allows the model to operate on smaller "chunks" of the data at a single instance, and hence on the full gamut of images - the size and overlap of these chunks can be customized by the user, (ii) lazy loading of the datasets themselves, allowing GaNDLF to only read the datasets into the memory during computation, and immediately deallocate the memory once it is used.

*Cross-validation.* k-fold cross validation[98,99] is a useful technique in ML that ensures reporting unbiased quantitative performance evaluation estimates of algorithmic generalizability on new datasets, i.e., by evaluating results on new unseen data discretized from an entire given data cohort. GaNDLF offers a nested k-fold cross-validation schema[100], where initially, cases of the complete cohort are randomly divided into k non-overlapping, equally-sized subsets and during each fold, k − 1 of these subsets are considered as the "retrospective"/"discovery" cohort and 1 as the "prospective"/"replication" cohort, which is unseen during training for this specific fold. Note that during each fold, the "prospective"/"replication" cohort is a different subset. This cross-validation scheme is analyzing the given data as if it had independent discovery and replication cohorts, but in a more statistically robust manner by randomly permuting across all given data. The number of folds for each level of split is specified in the configuration file, and the models for different folds can be trained in parallel (in accordance with the user's computation environment). GaNDLF also offers the option of specifying single fold training, if so desired.

*Zero/low-code principle.* The main entry point of GaNDLF's training mechanism follows a zero/low-code principle[101,102], where a dual file input is provided by the user, through the command line interface - a comma-separated-value (CSV) file and a text file (YAML) with intuitive indications of where to enter the training configuration parameters. The expected CSV file should comprise the subject identifiers along with the corresponding full paths of all required input images and masks (i.e., for segmentation workloads) and the values required for training and follow-up predictions (i.e., for regression and classification workloads). The subject identifiers are used to randomly split the entire dataset into training, validation, and testing subsets, using nested k-fold cross-validation[103]. The training can be configured to run on multiple DL accelerator cards, such as GPU or Gaudi. Furthermore, a YAML-based configuration file is used to control and parameterize all aspects of the training, such as the subject-based split of the cross-validation, data pre-processing, data augmentations (e.g., type, parameters, and probabilities), model parameters (e.g., architecture, list of classes, final convolution layer, optimizer type, loss function, number of epochs, scheduler, learning rate, batch size),

along with the training queue parameters (i.e., samples to extract per volume, maximum queue length, and number of threads to use). The YAML-based configuration file requires an indication of the GaNDLF version used to create the trained model, and the actual trained model, with the intention of ensuring coherence between these two.

*Monitoring & debugging.* GaNDLF also supports mixed precision training[104] to save computational resources and reduce training time. A single epoch comprises training the model using the training portion of the data and backpropagation of the generated loss, followed by evaluating the model performance on the validation portion of the data. In addition to saving the model trained after every epoch, each model corresponding to the best global losses for the training, validation, and testing datasets is also saved. These saved models can be used for subsequent inference, either using a single independent model or in a aggregated fashion utilizing label fusion[53,105]. Training statistics (such as the "Dice" similarity coefficient and loss) are stored for each epoch, for the training, validation and the testing data in the form of a CSV file, with the intention of facilitating simplified results reporting and detailed debugging.

*Handling class imbalance.* Class imbalance, i.e., where the presence of one class is significantly different in proportion to another, is a common problem in healthcare informatics[106,107]. To address this issue, GaNDLF allows the user to set a penalty for the loss function[108], which is inversely proportional to the classes being trained on. The penalty weights for the loss function will be defined as:

$$p_c = 1 - \frac{n_c}{N} \qquad (3)$$

where $p_c$ is the penalty for class 'c', and $n_c$ is the number of instances of the presence of class 'c' in the total number of samples N.

For example, for a classification workload using 100 cases, if there are 10 from class 0 and 90 from class 1, the weighted loss will get calculated to 0.9 for class 0 and 0.1 for class 1. This basically means that the misclassification penalty during loss back-propagation for class 0 (i.e., the "rarer" class) will be higher than that of class 1 (i.e., the more "common" class). The analogous process can be done for segmentation workloads as well. We recognize that this approach might not work for all problem types, and thus we have mechanisms for the user to specify a pre-determined loss penalty for greater customization.

**Inference mechanism.** GaNDLF's inference mechanism follows the same "zero/low-code" principle as its training mechanism, where the user needs a CSV file comprising the subjects' identifiers and the full paths of images, along with a YAML configuration file and the location of the trained models. For each trained model, the corresponding estimated output is stored and (depending on the user's parameterization) a final predicted output is generated by aggregating the outputs of the independent models. This aggregation happens through different approaches, subject to the prediction task, e.g., a label fusion approach may be used for segmentation workloads, averaging for regression workloads, and majority voting for classification. If the full paths of the ground truth labels are given in the input CSV, then the overall metrics (e.g., "Dice" and loss) of the model's performance are also calculated and stored.

*For radiology scans.* As soon as the data is read into memory, GaNDLF applies the pre-processing steps defined in the configuration file to each input dataset (see Section u for examples of these steps). Then TorchIO's[96] inference mechanism is used to enable patch-based inference for radiology images. This entails patch extraction, usually of the same size as the one that the corresponding model has been trained on, from the image(s) on which the model needs to infer. The forward pass of the model is then applied, and the result is stored in the corresponding location (Fig. 5a). This enables models to be trained and inferred on varied patch sizes based on the available hardware resources. Overlapping patches can be stitched by either cropping or taking an average of the predictions at the overlapping area, and the amount of overlap can be specified to ensure that dense inference can occur[96]. Although patch-based training and inference is being widely used, we note that various potential adverse effects of this process have been reported[109], requiring the operator's attention.

*For histology WSIs.* Histology WSIs need a different inference mechanism, than that for training, primarily due to their increased hardware requirements, i.e., WSIs can require more than 50GB when loaded completely on-memory. Fig. 5b illustrates this inference mechanism, which starts with the extraction of a WSI's imaging component at the maximum magnification/resolution (e.g., ×40) and its conversion to a TIFF with 9–10 layers of tiled images with different magnification levels (i.e., Fig. 5b(ii)-(iii) - "Data Fixing Pipeline"). The background area is then filtered out through the generation of a 'tissue mask' (Fig. 5b(iv)), using Red-Blue-Green (RGB) and Otsu-based thresholding[110,111], which is necessitated by the need to correctly tackle image reading issues occurring when trying to buffer any magnification level other than the lowest. This 'tissue mask' reduces the search space for downstream analyses, and hence reduces the overall computational footprint. This mask is further used to calculate foreground coordinates (Fig. 5b (v)), around which patches are extracted on-the-fly by leveraging TiffSlide's[112] dynamic read region property (Fig. 5b(vi)). This produces a

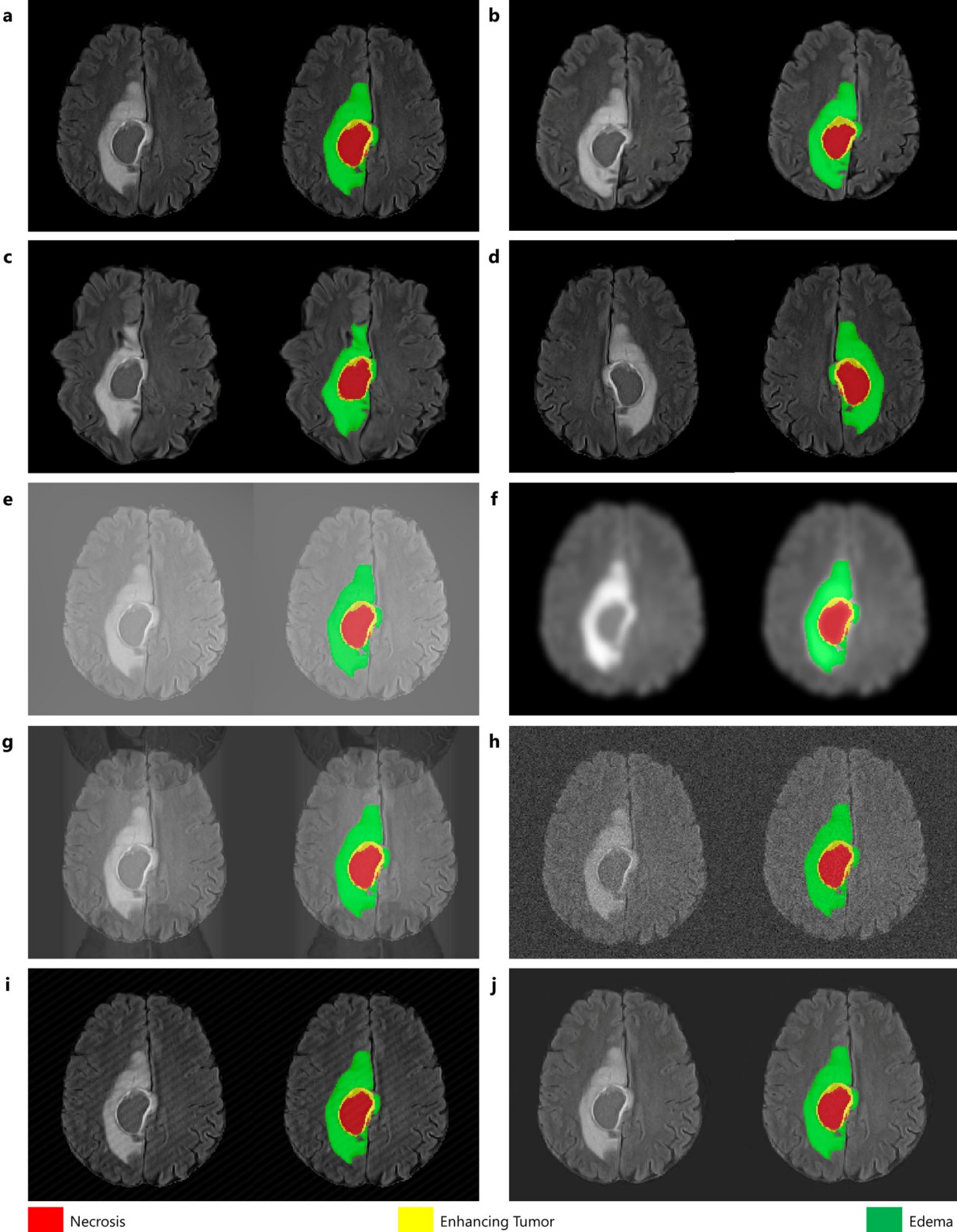

**Fig. 3 Illustration of various data augmentation techniques available in GaNDLF showing the image and overlaid segmentation. a** Original image,
**b** affine augmentation, **c** elastic augmentation, **d** flip augmentation, **e** bias augmentation, **f** blur augmentation, **g** ghosting augmentation, **h** noise
augmentation, **i** spike augmentation, and **j** motion augmentation.

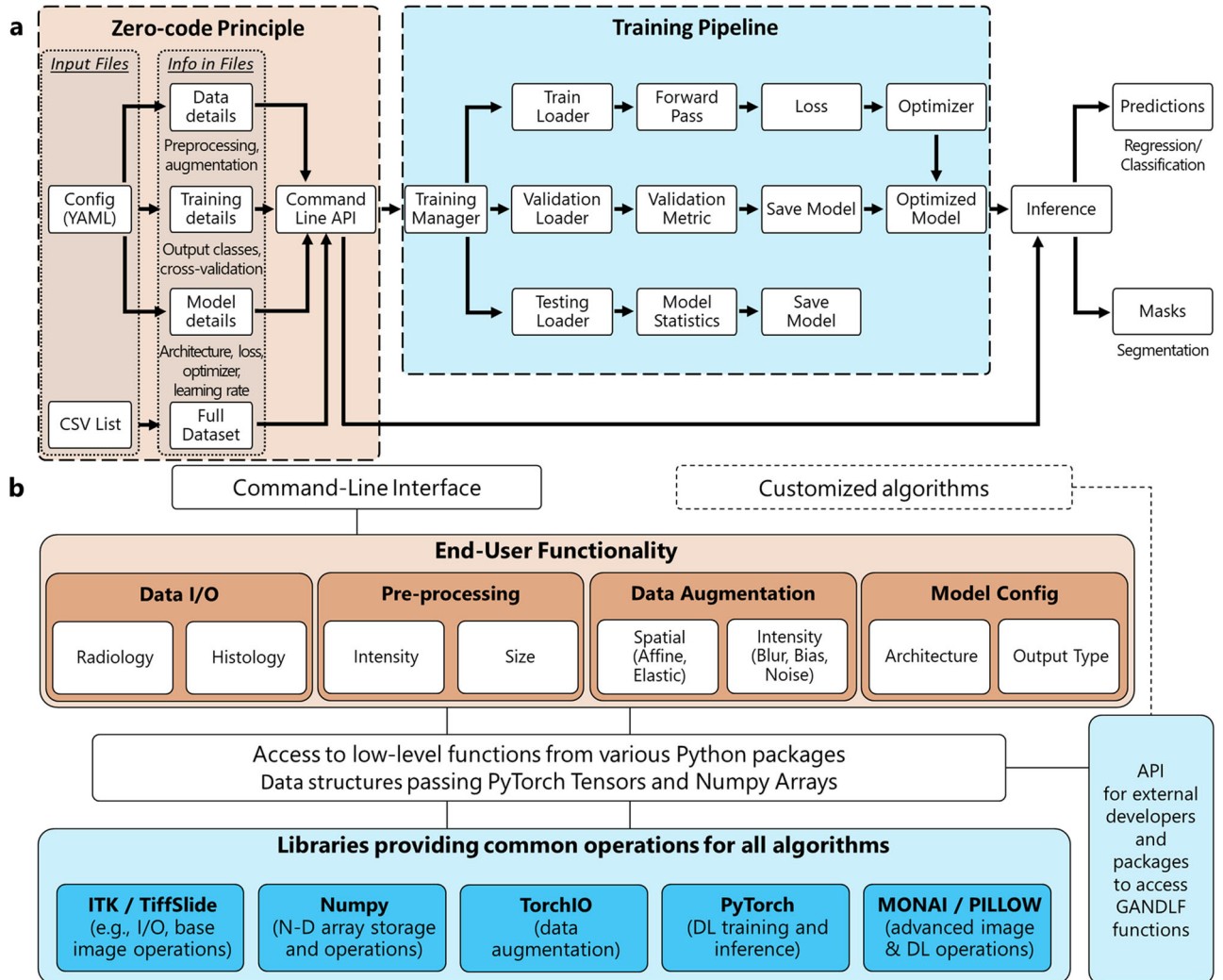

**Fig. 4 Overall structure of GaNDLF. a** Flowchart depicting the overall training procedure pipeline offered in GaNDLF. **b** GaNDLF 's software stack, highlighting the use of various low and high-level libraries to facilitate the creation of a flexible framework with an easy-to-use end-user interface. The orange pipeline represents the functionalities that can be changed using the "zero/low-code" principle, and blue pipeline represents the lower-level interactions of the codebase.

'count map' (Fig. 5b(viii)), which accounts for the contribution of overlapping patches for a tissue region ensuring probabilities are always between 0 and 1. The trained model is then used for a forward pass on each of these patches, producing an independent prediction for each. These predictions are then stitched together to form a 'segmentation probability map' (Figure 5b(ix)). The 'segmentation probability map' and the 'count map' are then multiplied to generate the 'final segmentation' output (Fig. 5b(x)).

**Post-processing**. It is conceivable that post-processing of a prediction would be required to get the most accurate result. GaNDLF provides a post-processing module that includes common image processing tasks, such as morphological operations (i.e., dilation, erosion, closing, opening), and the ability to map predicted labels from one value to another. The former is useful in cases where segmentation predictions are generated with holes and need to be closed, and the latter can be used to assign the desired final label values to a prediction.

**Modularity and extendibility**. A description of GaNDLF's software stack, modularity, and extendibility is hereby provided, as well as how the lower-level libraries are utilized to create an abstract user interface, which can be customized based on the application at hand. Following this, the flexibility of the framework from a technical point-of-view is chronicled, which illustrates the ease with which new functionality can be added. Further details on customizing the entire processing pipeline (including hyper-parameter tuning and optimization) can be found in the software documentation at: mlcommons.github.io/GaNDLF.

*Software Stack*. The software stack of GaNDLF, illustrated in Figure 4b, depicts the interconnections between the lower level libraries and more abstract functionalities exposed to the user via the command line interface. This ensures that a researcher can perform DL training and/or inference without having to write a single line of code. Furthermore, the flexibility of the stack is demonstrated by the ease with which a new component (e.g., a pre-processing step, or a new network architecture) can be incorporated into the framework, and subsequently applied to new types of data/applications with minimal effort. Specifically, the framework's flexibility affects components listed in the following subsections.

*Dimensions*. To ensure maximum flexibility and applicability across various types of data, GaNDLF supports both 3D and 2D datasets. Using the same codebase, GaNDLF has the ability to apply various architectures across diverse modalities such as MRI, CT, retinal, and digitized histology WSI, including immunohistochemical (IHC), In Situ Hybridization (ISH), and H&E stained tissue sections.

*Input channels/modalities & output classes*. GaNDLF supports multiple input channels/modalities/sequences and output classes, for either segmentation, classification, or regression, to ensure maximal applicability across various problem domains, whether it involves a binary task (e.g., brain extraction) or multi-class outputs (e.g., brain tumor sub-region segmentation).

- Radiology images require the ability to process both 2D and 3D data. Although imaging examples that GaNDLF has been applied and evaluated so far describe CT, MRI, and tomosynthesis scans, it offers support for almost every radiology image via ITK.

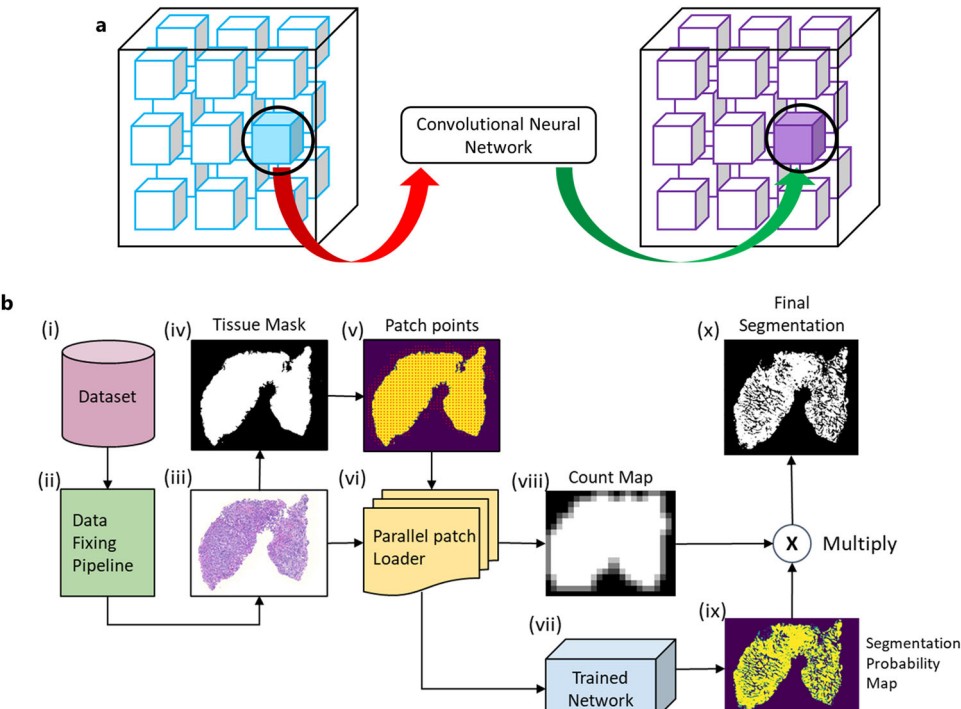

**Fig. 5 GaNDLF's inference mechanism for medical images. a** Illustration of the patch-based inference mechanism for radiology images. This process is repeated till the complete image gets processed. **b** Steps to perform specialized inference for histology images. Starting with the raw whole slide image (WSI), multiple specialized pre-processing steps (ii-vi) are performed before a patch can be given as input to a trained model. The coordinates of each patch need to be saved along with the overlap information in order to obtain the final result.

- Histology images, on the contrary, require specialized handling along the following criteria:

  - Input: The use of TiffSlide[112] allows GaNDLF to read a fraction of the entire WSI data at the resolution closest to the requested magnification level, thereby ensuring memory-efficiency.
  - Patch-extraction: Since a WSI cannot always be processed on its entirety due to hardware constraints, a patch-based mechanism considering multiple resolutions is essential. This mechanism is offered through our open-source Open Patch Miner (OPM, github.com/CBICA/OPM), which has been integrated within GaNDLF for simple and rapid batch-processing of patches. OPM can automatically mask tissue in a WSI and convert the highest available resolution to square patches, given a pre-defined overlap amount and patch dimensions. Specifically, it extracts patches with the pre-defined overlap using a pseudo-grid and parallel sampling adjustable for tissue inclusion, in proportion to different tissue classes (for classification workloads), and while omitting the background region.

*Network architectures.* GaNDLF seeks to provide both well-established and state-of-the-art network architectures showing promise in the field of healthcare. The currently available (and ever expanding) architectures offered through GaNDLF, and their detailed descriptions are provided in the Supplementary Methods: Network Architectures as well as their illustrations in Supplementary Figs. 1–10.

*Applications.* As previously stated, GaNDLF can train DL models to target various workloads, including segmentation, regression, and classification. Depending on available resources, most models can be extended for all these workloads (such as UNet), and there are workload-specific models, such as the brain age prediction model[51], which modifies a VGG-16 model pre-trained on ImageNet weights and is only defined for regression. The flexibility of GaNDLF's framework makes it possible for all these models to co-exist and to leverage the robustly designed data loading and augmentation mechanisms for future study extensions. Having a common API for all these workloads also makes it relatively easy for researchers to start applying well-defined network architectures towards various problems and datasets, thereby contributing in getting DL-based pipelines into clinical workflows.

*Performance evaluation.* We provide different options to evaluate the model performance during training, and mechanisms to incorporate new validated recommendations[113] as needed. Below definitions of the metrics used in the results section of this manuscript are provided. Specifically, for segmentation workloads,

the "Dice Similarity Coefficient"[114] (Eq. (4)) is mostly used as the performance evaluation metric, and all related models were trained to maximize it. "Dice" is a common metric used to evaluate the performance of segmentation workloads. It measures the extent of spatial overlap, while taking into account the intersection between the predicted masks ($PM$) and the provided ground truth ($GT$), hence handles over- and under-segmentation.

$$Dice = \frac{2|GT \cap PM|}{|GT| + |PM|} \quad (4)$$

Additionally, the "Hausdorff Distance"[115] is a metric for segmentation workloads (Eq. (5)). This metric quantifies the distance between the boundaries of the ground truth labels against the predicted label. It is sensitive to local differences, as opposed to "Dice", which represents a global measure of overlap.

$$H_{95}(PM, GT) = \max \left\{ P_{95\%} \underset{p \in PM}{d(p, GT)}, P_{95\%} \underset{g \in GT}{d(g, PM)} \right\} \quad (5)$$

where $d(x, Y) = \min_{y \in Y} ||x - y||$ is the distance of $x$ to set $Y$.

For regression workloads, we used the Mean Squared Error ("MSE")[116] as our evaluation metric and all models were trained to minimize it. "MSE" measures the statistical difference between the target prediction $T$ and the output of the model $P$ for the entire sample size $n$ (illustrated by Eq. (6)). The same mechanism has been used for accuracy, macro-averaged F1-score, and area-under-the-curve, among others by leveraging TorchMetrics[117].

$$MSE = \frac{1}{n} \sum_{i=1}^{n} (T_i - P_i)^2 \quad (6)$$

For classification workloads, we used the balanced accuracy ("Acc")[118] as an evaluation metric and trained models to minimize the cross entropy loss[119]. "Acc" can be used for both binary and multi-class classification, and is defined as the arithmetic mean of sensitivity and specificity. This metric is especially useful when dealing with imbalanced data, i.e. when one of the target classes appears a lot more than the other[118].

$$Acc = \frac{\left(\frac{TP}{TP+FN}\right) + \left(\frac{TN}{TN+FP}\right)}{2} \quad (7)$$

where $TP$ & $TN$ are the number of true and false positives, and $FP$ & $FN$ are the number of false positives and negatives, respectively.

**Interpretability tools.** It is an ongoing problem that deep neural networks lack the interpretability or explainability necessary for medical practitioners to trust into the

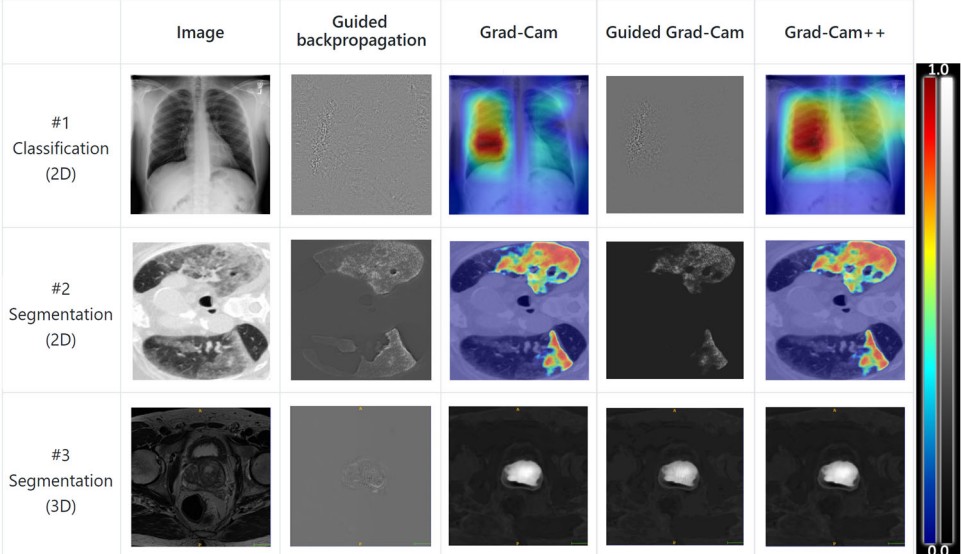

**Fig. 6 Examples of generated M3D-CAM attention maps[56] with the Grad-CAM backend.** The top row of images show the attention from a 2D classification network, the middle row from a 2D segmentation network and the bottom row from a 3D segmentation network.

networks decisions, hindering the practical application of such models in clinical practice[120,121]. To counter this, GaNDLF integrated the PyTorch library M3D-CAM[56], which enables the easy generation of attention maps of CNN-based models for both 2D and 3D data, and is applicable to both classification and segmentation models (examples illustrated in Fig. 6). The attention maps can be generated with multiple methods: Guided Back-propagation[122], Grad-CAM[123], Guided Grad-CAM[123] and Grad-CAM++[124]. The maps visualize the regions in the input data that most heavily influenced the model prediction at a certain layer.

**Model optimization**. Typical clinical environments do not have access to specialized hardware (such as DL acceleration cards) and increased memory, which are necessary for practical on-premise deployment of DL models. This situation is further exacerbated in developing regions, where clinical environments are even more limited in resources. The question of training, or even inference/execution, of DL models has not received sufficient attention in current literature, hindering clinical translation of state-of-the-art models. One of the reasons that this clinical applicability is typically not considered during the life-cycle of a research project is because of the added complexity. Thus, to further the use of models trained using GaNDLF in clinical/low-resource settings, GaNDLF incorporates post-training optimization of all models using OpenVINO[125] by default, and provides the optimized model as an additional output at the conclusion of any model training procedure. This enables inference of DL models to be deployed to low-resource machines[126], which magnifies their impact in precision medicine.

**Software development practices**. GaNDLF incorporates several well-known robust software development measures[127] to ensure ongoing software quality in the presence of community contributions. These measures include the following:

- "Unit testing" refers to tests of individual functional components of the software, to ensure that implementation changes do not break the usage contract established by that component. These units are the smallest relevant units of functionality, and testing these helps ensure that bugfixes, feature additions, and performance optimizations do not cause breaking changes to basic calculations made by the software, such as those that would impact model training. GaNDLF includes extensive unit tests for all custom functionality which is built atop other libraries.
- "System testing" refers to larger-scale tests of software functionality, to test the usage of the software in a broader way that more closely correlates to real usage. GaNDLF's test suite includes extensive system tests, including tests of each major usage mode (training, inference, data preparation, and so on), and tests for each model architecture across types of data (such as radiology and histology images) and types of workloads (such as classification, regression, and segmentation) as appropriate. GaNDLF's test suite requires all tests to pass before code can be committed to the repository, and changes cannot be committed to the code repository if any tests fail for any reason.
- Automated and publicly-declared vulnerability testing of code dependencies via Dependabot[128], which ensures that GaNDLF stays up-to-date on security patches.
- "Automated test coverage reports" are a metric collected during testing, reflecting how much of the codebase is traversed by tests. Higher code

coverage indicates that more individual components, functions, and conditional branches of the software have been tested. GaNDLF automatically reports code coverage changes on any incoming contribution and flags changes that decrease code coverage for further review.
- "Continuous deployment" via containerization using the Docker, Singularity, and MLCube standards.

While the above tests cover code-level reliability, it is difficult to infer reliability regarding performance of the models produced by GaNDLF, in part due to stochasticity of the training process. We are actively working on additions to the automated test suite that would measure performance of each model on small sample datasets, and flag contributions that cause drops in performance for further review.

**Experimental design**. For each application, multiple models are trained in accordance with the cross-validation schema described in Methods Section. For performance evaluation, we use the model with the best validation score as defined in the application-specific evaluation criteria and apply this model to the test dataset for each fold, giving us the average performance of an architecture for the specific problem. To maintain reproducibility and prevent overfitting, we have trained each architecture with a 20/16/64 split, which results in the training of 25 models in total, for each architecture. Specifically, the 20/16/64 split comprises 5 non-overlapping splits (i.e., each containing 20%) of the complete dataset. Each one of these splits is set aside as the testing cohort for each fold. From the remaining 80% of the complete data during this fold, 5 further splits are done, each containing 16% of the full data, and used for validation. Finally, the remaining data for this fold, which represent 64% of the full cohort, are used for training.

*Segmentation of brain in MRI*. Brain extraction is an essential pre-processing step in the realm of neuroimaging, and has an immediate impact on the quality of all subsequent processing and analyses steps. We have used a multi-institutional dataset of 2520 MRI scans along with their corresponding manually annotated brain masks. We trained on 1320 scans in a modality-agnostic manner (i.e., each structural MRI scan was treated as a separate input) as described in ref. [11] and setting a internal validation set of 180 scans, with an independent testing cohort of 360 scans to ascertain the model performance. We trained by resampling the data from an isotropic resolution of 1 mm$^3$ with a shape of $240 \times 240 \times 160$ to a anisotropic resolution of $1.825 \times 1.825 \times 1.25$ mm$^3$ with a shape of $128 \times 128 \times 128$[11]. The reason for this resampling was GPU memory limitations, i.e., 11GB VRAM. We trained multiple architectures (UNet, ResUNet, FCN) with only z-score normalization by discarding the zero-voxels, with no augmentations enabled.

*Segmentation of brain tumor sub-regions in MRI*. Gliomas are among the most common and aggressive brain malignancies and accurate delineation of these regions can provide valuable clinical insights. We have used the publicly available MRI data from the International Brain Tumor Segmentation (BraTS) challenge of 2020[7–10,129,130] to train multiple models to segment the various brain tumor sub-regions. Specifically, we used the full cohort of 371 training subjects, which we iteratively split it into 74 testing, 60 validation, and 237 training subjects following the k-fold cross-validation schema mentioned in the Cross-Validation sub-section in Methods, with all the 4 structural MRI sequences making up a single input

data-point[11]. In total, 25 models are trained for each architecture (UNet, ResUNet, UInc, and FCN). For each model, we used a set of common hyperparameters that runs in a GPU with 11GB of memory, namely, patch size of $128 \times 128 \times 128$, 30 base filters, "Dice" loss, with stochastic gradient descent (SGD) as the optimizer. For pre-processing, we used z-score normalization by discarding the zero-voxels and cropping of the zero-planes. For data augmentation, we used noise, flipping, affine, rotation and blur, each with a probability of getting picked as 0.35. In each case, the model is trained to maximize the performance evaluation criteria, which is constructed by following the instructions in the BraTS challenge[7,10], i.e., averaging the "Dice" across the enhancing tumor, the tumor core (formed by combining necrosis and enhancing tumor), and the whole tumor (formed by combining the tumor core and the peritumoral edematous/infiltrated tissue).

*Whole brain parcellation in MRI.* Whole brain structural segmentation could provide richer neuroanatomy information in neuroimaging studies where those structures are relatively small and thus it becomes a more challenging task to accurately segment them in the similar image appearances[45]. We have used the publicly available MRI data from the Multi-Atlas Labelling Challenge (MALC) of 2012 to train multiple models to segment the whole brain into 133 fine-grained sub-regions[131] from T1 weighted scans. Specifically, this challenge dataset contains in total of 30 scans, where a training list of 15 scans and a testing list of 15 scans are provided from the challenge. We trained by resampling the data into an isotropic resolution of 1 mm$^3$ with a shape of $256 \times 256 \times 256$ as referred from prior work[45]. Particularly, a ResUNet model and an UNet model are implemented for training with a set of common hyperparameters that runs in a GPU with 24 GB of memory, 30 base filters, Dice loss, with SGD as the optimizer. Differently, ResUNet used a learning rate of 0.02 and a patch size of $64 \times 64 \times 64$, whereas UNet used a learning rate of 0.01 and a patch size of $96 \times 96 \times 96$. For pre-processing, the dataset are normalized into range of $[-1, 1]$[45] with no augmentations enabled.

*Segmentation of fatty and dense breast tissue using DBT.* Breast density has been widely demonstrated to be an independent risk factor for breast cancer[132–134]. Given the rise of digital breast tomosynthesis (DBT) in breast cancer screening compared to traditional 2D mammography[46], there is potential to estimate volumetric breast density (VBD) routinely using machine learning methods. We retrospectively analyzed 1080 negative DBT screening exams completed between 2011 and 2016 at the Hospital of the University of Pennsylvania that contained both 2D raw DBT and 3D reconstructed images. Using the available cranio-caudal and mediolateral-oblique views for each patient, a total of 7850 DBT views were available. We created a convolutional neural network that employed the U-Net architecture for a 3-label image segmentation problem (background, fatty breast tissue, dense breast tissue). Training, validation, and testing sets comprised 70%, 15%, and 15% of the original dataset, respectively. Corresponding ground truth segmentations were generated from a previously validated software that generated VBD metrics based on both 3D reconstructed slices and raw 2D DBT data. 24 models were trained, each using a unique combination of learning rates, batch sizes, patch sizes, and optimizers. Data augmentation during training included affine transformations, blur transformations, and noise transformations, with probabilities of 0.25, 0.5, and 0.5, respectively. The performance of each model was based on weighted and unweighted Dice scores and the final model was selected based on validation set performance.

*Segmentation of structural tumor volume from breast MRI.* The ACRIN 6657/I-SPY1 TRIAL[47,135] enrolled 237 women from May 2002 to March 2006. From these cases, after applying the inclusion/exclusion criteria, we were left with 163 subjects which contained the 3 time-points of interest with regards to contrast injection. These were pre-injection, and 2 post-injection scans. The first-post contrast image for each case was used by the radiologist to delineate the entire 3-D primary tumor segmentation for each patient, also known as the "structural tumor volume", since it contained peak excitation of the contrast agent[47,135]. We trained the ResUNet using all the 3 time-points using an initial and minimum learning rates of 0.01 and $10^{-4}$, respectively, driven using the SGD optimizer. We observed an average "Dice" of 0.74 across 5 fold cross-validation[136].

*Segmentation of lung field in CT.* An accurate volumetric estimation of the lung field would be crucial towards furthering the clinical goals of tackling respiratory illnesses, such as influenza, pneumonia, and COVID-19 pathologies. However, manual segmentation of the lung field is time-intensive and subjective with low inter-individual reliability, especially for large-scale datasets. Automatic segmentation algorithms can substantially accelerate the analytical procedure. We trained 3D lung field segmentation models with two internal datasets from two independent cohorts based on the ResUNet structure. The first dataset was identified within the lung cancer screening cohort at the University of Pennsylvania Health System (UPHS), and consisted of 500 low-dose CT scans in which 25 were diagnosed with lung cancer. Their corresponding ground truth segmentations for the lung field were generated under a semi-automatic procedure leveraging 2-cluster k-means, followed by manual qualitative refinements. The second dataset contains 673 low-dose CT scans identified within COVID-19 patients admitted to UPHS. Because of the difficulties posed by pathological presentations of COVID-19 in scans, the ground truth was obtained by manually choosing scans with correct

segmentations generated by the algorithm that worked on individual slices and accounted for the presence of severe pathologies[137]. We trained our models on the two datasets separately. We split both datasets into training, validation and test sets. For the first dataset, there are 254 scans in the training set, 64 scans in the validation set and 182 scans in the test set. For the second dataset, there are 360 scans in the training set, 98 scans in the validation set and 215 scans in the test set. We performed windowed pre-processing and clipped the intensities between $[-900, -300]$ Hounsfield Units (HU). We also resampled the data down to $[128 \times 128 \times 128]$ in order to consider the entire chest region and to ensure that the trained model remained agnostic to the original image resolution. We trained the ResUNet architecture with clipping and z-score normalization by discarding the zero-voxels with no augmentations enabled. The "Dice" score was employed as our evaluation metric and the model was trained to maximize the "Dice" score.

*Segmentation of retinal fundus.* We used the dataset from the PALM challenge[48], which consists of segmentation of lesions in retinal fundus images and replicated the results for a ResUNet architecture from[138]. Additionally, we trained on FCN, UNet, and UInc to show results on a diversified set of architectures from the same dataset. We used the full cohort of 400 training subjects, and iteratively split into 80 testing, 64 validation, and 256 training subjects following the k-fold cross-validation schema mentioned in the Cross-Validation sub-section in Methods. In total, 25 models are trained for each architecture (UNet, ResUNet, UInc, and FCN). For each model, we used a set of common hyperparameters options that runs in a GPU with 11GB of memory, namely, patch size of $2048 \times 1024$, 30 base filters, "Dice" loss, with SGD as the optimizer. For pre-processing, we used full-image normalization, and data augmentation was performed using flipping, rotation, noise and blur, each with a probability of 0.5. The performance is evaluated in comparison with the ground truth binary masks of the fundus in the testing set.

*Segmentation of quadrants in panoramic dental X-ray images.* Dental enumeration from panoramic dental X-Ray images has a crucial role in the identification of dental diseases. Performing that task with deep learning provides an extensive advantage for the clinician to number the dentition quickly and point out the teeth that need care more accurately. Quadrant segmentation from those panoramic images is the first and the most critical step of numbering the dentition accurately, and a previous study has used an UNet model to achieve that task[49]. Here, we replicated those results by training a segmentation model with GaNDLF that extracts quadrants from the dental X-ray images. To do that, we have used 900 dental X-ray images with their corresponding five classes (one for each quadrant plus the background). Class annotations have been generated by the experts and the images were resized down to $128 \times 128$ in order to consider the entire mouth region. We trained the UNet, the ResUNet, and the FCN architectures with 30 base filters with z-score normalization with no augmentations enabled. We used "Dice" as the evaluation metric and trained the model to maximize it.

*Segmentation of colorectal cancer in WSI.* Colonoscopy pathology examination can find cells of early-stage colon tumor from small tissue slices, and pathologists need to examine hundreds of tissue slices on a day-to-day basis, which is an extremely time consuming and tedious work. The DigestPath challenge[50] motivated participants to automate this process and thereby contribute to potentially improved diagnostics. The data provided in the DigestPath challenge includes slides containing colorectal cancer in JPEG format. The dimensions of the provided images range from $3000 \times 3000$ to $30000 \times 30000$. 180000 patches of the shape $512 \times 512$ at 10× resolution were extracted for training and 30000 for validation, with a set of 30 WSIs being kept separate as independent testing dataset. We trained the ResUNet architecture, and prior to training we normalized the training values to $[0-1]$ by dividing each pixel by the maximum possible intensity, i.e., 255. To account for model generalizability, we employed the flip, rotate, blur, noise, gamma, and brightness data augmentations. We used "Dice" as our evaluation metric and trained the model to maximize it. Inference was then done on the testing dataset and the output of the model was evaluated against the ground truth binary masks to calculate the "Dice" score.

*Brain age prediction from MRI.* The human brain ages differently because of various environment factors. Quantifying the difference between actual age and predicted age can provide a useful insight into the overlap of aging signatures with various neurodegenerative pathologies. A 2020 study[51] has used common 2D CNN architectures, borrowed from the computer vision community, to predict brain age from T1-weighted MRI scans across a wide age range. Methodologically, the original fully connected layers of the VGG-Net was replaced by a global average pooling, followed by a new fully connected layer of size 1024 with 80% dropout, and then a single output node with a linear activation was added. The network was then trained with the Adam optimizer[139], while using MSE. This study was evaluated on 10,000 diverse structural brain MRI scans, pooling data from various studies, including the UK Biobank[140] and a multisite schizophrenia consortium[141], thereby representing various subject populations and acquisition protocols. This inherent variability of the collective dataset allowed to successfully learn a regression model generalizable across sites. The study in question study[51] goes on to examine using the learned age

prediction weights as a starting point for transfer learning to other neuroimaging workflows. It is shown that the age prediction weights serve as a superior basis for transfer learning compared to ImageNet, particularly in neuroimaging problems, where the new training data is limited[51].

Leveraging the modular nature of GaNDLF, we replicated the age prediction results of that study[51] using the same model architecture, training schema, and dataset as in the original study, while following GaNDLF's procedures. Using the VGG-16 model architecture and GaNDLF's built-in cross-validation functionality, we trained regression models using the intermediate 80 axial slices of each subject, with input data being split on the subject level. The same network hyperparameters were used, as those specified in the original study[51].

*Classification of diabetic foot ulcer images.* Diabetic foot ulceration (DFU) is a serious complication of diabetes, which poses a major problem for health systems around the world. DFU can further lead to infection and ischemia, which can result in the amputation of limbs, with more severe cases being terminal illnesses. Diabetic Foot Ulcers Grand Challenge (DFUC) 2021[52] is conducted to help early detection of DFU, which can prevent turning into more serious cases, improve care and reduce the burden on healthcare systems.

DFUC 2021 required its participants to solve a multi-class classification problem through DFU images. The dataset contain DFU images of 4 different classes, labelled as 1) "infection", 2) "ischemia", 3) "both infection and ischemia", and 4) "controls" (i.e., neither infection, nor ischemia). The original resolution of the images are $224 \times 224$. The dataset consist of 15,863 images, partitioned into three distinct independent subsets. The training set includes 5955 images, where 2555 cases with only "infection", 227 cases with only "ischemia", 621 cases with "both infection and ischemia", and 2552 "control" cases.

Through the challenge, ease of conducting different experiments through GaNDLF assisted us to train well-known generalizable models. Three different versions of the VGG architecture[142,143] and one version of DenseNet architecture are experimented with GaNDLF, namely VGG11, VGG16, VGG19 and DenseNet121. We utilized *k*-fold cross validation functionality of GaNDLF to prevent overfitting. We applied patching with size of $128 \times 128$. We set the batch size as $b = 256$ for VGG11 and VGG16 architectures, $b = 128$ for VGG19, $b = 32$ for DenseNet121 (high GPU resources were not available for this experiment) to ensure maximal utilization of the available hardware resources. Adding bias, blur, noise, and swapping techniques are used as data augmentation with probability $p = 0.5$. Z-scoring normalization is used for data pre-processing. Cross-entropy loss is used as the loss function, which is shown work well for multi-class classification problems. We also experimented weighted cross entropy loss[144], which generally works better for imbalanced classes. The Adam optimizer[139] was used with an initial learning rate of $lr = 0.001$.

*Classification of TILs using histology scans.* Electronic capture (digitisation) and analyses of whole slide images (WSIs) of tissue specimens are becoming ubiquitous. Digital Pathology interpretation is becoming increasingly common, where many sites are actively scanning archived glass tissue slides with commercially available high-speed scanners to generate high-resolution gigapixel WSIs. Alongside these efforts, a great variety of AI algorithms have been developed to extract many salient tissue and tumor characteristics from WSIs. Examples include segmentation of tumor regions, histologic subtypes of tumors, microanatomic tissue compartments; detection and classification of immune cells to identify tumor-infiltrating lymphocytes (TILs); and the detection and classification of cells and nuclei. TILs are lymphoplasmacytic cells that are spatially located in tumor regions, where their role as an important biomarker for the prediction of clinical outcomes in cancer patients is becoming increasingly recognised[145–147]. Identification of the abundance and the patterns of spatial distribution of TILs in WSI represent a quantitative approach to characterizing important tumor immune interactions. We created a cohort of pre-defined training and validation cases consisting of patches extracted from WSIs of cancer from 12 anatomically distinct sites, comprising of breast, cervix, colon, lung, pancreas, prostate, rectum, skin, stomach, uterus, and uvea of the eye. All cases are publicly available in The Cancer Genome Atlas (TCGA)[148].

We have used a VGG-16 architecture[142,143] that has been pretrained using the ImageNet dataset[55]. We updated the architecture's first and final layers to be able to process input images of any size, and only output the 2 relevant classes for this problem, respectively[54]. We then proceed with training this architecture using different schedulers and optimizers along with varying learning rates to get an average performance of 0.89. The best results were seen with the step scheduler on Adam optimizer[139] using a learning rate of 0.001.

*Prediction of EGFRvIII using structural MRI.* Glioblastoma (GBM) is the most common and aggressive primary malignant adult brain tumor and epidermal growth factor receptor variant III (EGFRvIII) mutation has been considered a driver mutation and therapeutic target in GBM[149–151]. Usually, EGFRvIII presence is determined by analysis of surgically resected or biopsy-obtained tissue specimens. We are conducting experiments towards prediction of the EGFRvIII status non-invasively, by analyzing the preoperative and pre-processed structural multiparametric (mp)MRI sequences (T1, T2, T1-Gd and T2-Flair). We identified a cohort of 146 patients containing these four scans acquired at the Hospital of the University of Pennsylvania.

We trained the VGG11 classification architecture utilizing the k-fold cross validation functionality to classify the EGFRvIII status as positive or negative based on the four structural modalities as well as the segmentation map of tumor core. The patch size was set to $128 \times 150 \times 131$ the various experiments were carried out to find the optimal set of hyperparameters utilizing the various options available in GaNDLF. Baseline results were obtained without using any additional data augmentation techniques. Best performance was achieved with cross entropy loss function, SGD optimizer and step scheduler with learning rate of 0.1.

**Reporting summary**. Further information on research design is available in the Nature Portfolio Reporting Summary linked to this article.

## Data availability

The data used for each of the experiments are available as follows:

Segmentation of Brain in MRI: The data used was a combination of a publicly available dataset[8,11], augmented with scans from private collections of multiple institutions, namely the University of Pennsylvania Health System (UPHS), Thomas Jefferson University, and MD Anderson Cancer Center. The data that support the findings of this study are available from the individual hospitals, but restrictions apply to the availability of these data, which were used under license for the current study, and so are not publicly available. Data are however available from the authors upon reasonable request and with permission of the aforementioned clinical sites.

Segmentation of Brain Tumor Sub-regions in MRI: The data used was from the Brain Tumor Segmentation (BraTS) challenge of 2020[7–10].

Whole Brain Parcellation in MRI: The data used was from the Multi-Atlas Labelling challenge (MALC) of 2012[131].

Segmentation of Breast Tissue using DBT: The data that support the findings of this study are available from the UPHS, but restrictions apply to the availability of these data, which were used under license for the current study, and so are not publicly available. Data are however available from the authors upon reasonable request and with permission of the University of Pennsylvania.

Segmentation of Structural Tumor Volume Breast MRI: The data used in this study was obtained from the ACRIN 6657/I-SPY1 TRIAL[47,135] and can be downloaded from https://wiki.cancerimagingarchive.net/display/Public/ISPY1.

Segmentation of Lung Field in CT: The data that support the findings of this study are available from the UPHS, but restrictions apply to the availability of these data, which were used under license for the current study, and so are not publicly available. Data are however available from the authors upon reasonable request and with permission of the University of Pennsylvania.

Segmentation of Retinal Fundus: The data used was from the PALM challenge[48].

Segmentation of Quadrants in Panoramic Dental X-Ray Images: The data that support the findings of this study are available from the Istanbul Medipol University, but restrictions apply to the availability of these data, which were used under license for the current study, and so are not publicly available. Data are however available from the authors upon reasonable request and with permission of the Istanbul Medipol University.

Segmentation of Colorectal Cancer in WSI: The data used was from the DigestPath challenge[50].

Brain Age Prediction from MRI: The data used was from the UK Biobank[140] and a multisite schizophrenia consortium[141].

Prediction of the EGFRvIII mutation in brain tumors using structural mpMRI: The data that support the findings of this study are available from the UPHS, but restrictions apply to the availability of these data, which were used under license for the current study, and so are not publicly available. Data are however available from the authors upon reasonable request and with permission of the University of Pennsylvania.

Classification of Diabetic Foot Ulcer Images: The data used was from the Diabetic Foot Ulcer Grand Challenge (DFUC) of 2021[52].

Classification of Tumor Infiltrating Lymphocytes: The data used is available in The Cancer Genome Atlas (TCGA)[148].

## Code availability

To encourage reproducibility, all the code used for this work is open-sourced at github.com/mlcommons/GaNDLF, and it can be installed as detailed in mlcommons.github.io/GaNDLF/setup.

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

## Acknowledgements

GaNDLF is now primarily maintained and supported by MLCommons (mlcommons.org). Research reported in this publication was partly supported by the National Cancer Institute (NCI), the National Institute of Neurological Disorders and Stroke (NINDS), the National Institute on Aging (NIA), the National Institute of Mental Health (NIMH), and the National Institute of Biomedical Imaging and Bioengineering (NIBIB) of the National Institutes of Health (NIH), under award numbers NCI:U01CA242871, NCI:U24CA189523, NINDS:R01NS042645, NIA:RF1AG054409, NIA:U01AG068057, NIMH:R01MH112070, NCI:R01CA161749 and NIBIB:R01EB022573. S.A. Tsaftaris acknowledges the support of Canon Medical and the Royal Academy of Engineering and the Research Chairs and Senior Research Fellowships scheme (grant RCSRF1819\8\25). B.M. acknowledges support by the Helmut-Horten-Foundation. A.K. receives funding from IHU Strasbourg under award number ANR-10-IAHU-02. The content of this publication is solely the responsibility of the authors and does not represent the official views of the NIH or any other funding body.

## Author contributions

Idea Conception: S.P., S.P.T., M.S., A.K., R.U., P.M., S.B. Development of software: S.P, S.P.T, I.E.H, U.B., B.B., Me.B., O.G., S.M., S.T., K.G., Cam.G., Cal.G., A.G., B.E., M.S., J.W., D.K., R.P. Data Acquisition, Processing, and Analysis: S.P, D.L., V.A., C.Z., V.B., Y.L., B.H., R.C., S.A., T.M.K., J.H.S., Y.F., A.G., S.E., S.B. Review and Edit of manuscript:

A.G., M.B., J.H.S., Y.F., P.S., A.M., S.A.T., B.M., C.D., D.K., A.K., R.U., P.M., S.B. Writing the Original Manuscript: S.P., S.P.T., U.B., B.B., S.B. Review, Edit, & Approval of the Final Manuscript: All authors.

## Competing interests

The authors declare no Competing Interests.
