## [Peer Review File · Communications Engineering]

Reviewers' comments:

Reviewer #1 (Remarks to the Author):

This paper and its corresponding software framework mainly aim to develop an 'All-in-One' Pytorch-based python package that contains everything needed to perform a streamlined deep learning training with - Data loader; Built-in module for data preprocessing and data augmentation for common medical imaging classification and segmentation tasks; a collection of well-established (however lacking some of latest ones) neural network models, e.g. UNet, ResNet, etc; a universal predefined training workflow that can be controlled by simply edit the configuration file; logging mechanism to monitor the training process; included some capability for explainable AI, e.g. Grad-Cam

It is mainly designed for easy use of DL to train and test DL models in a clinical environment with low-programming requirements. This is however very similar to other frameworks, e.g. Monai. Essentially, this framework collected the code from other open source projects and put it all together and linked them with an abstracted API layer.

Apart from the above, please find below detailed comments:

1. Currently, the models contain a limited number of models, which are not up to date. There are no transformer-based models included in the framework. Although the framework is designed to be easy to extend, however, this highly relies on the maintainers regarding the quality of the implementation.
2. Zero- and Low- code solution is targeting a small group of clinical audience, and provides a 'click-to-run' way to try out some deep learning models. However, there is a lack of a mechanism for optimizing the hyper-parameters which are the keys to performance.
3. Not support multiple modality inputs. Currently, it seems only supports single inputs (with multi-channel support).
4. Only support a very standard supervised training pipeline. Many update-to-date methods, inc. GAN, self-supervised, etc are not supported.
5. Similarly, it does not support iterative DL, etc.
6. It is not clear how the framework handles large datasets, particularly imaging data in 3D, in an efficient way, given what they mentioned the main usage of this framework might be in resource-constrained environments.
7. There is a lack of distributed and parallel training support, which limits the usage in real environments with relatively large datasets, 3D models and large Transformer-based models.

Reviewer #2 (Remarks to the Author):

This manuscript proposes an open-source framework named Generally Nuanced Deep Learning Framework (GaNDLF) for end-to-end deep learning (DL) based medical image analysis applications. It's a general framework which could process both radiology and histology data for a variety of DL workloads/tasks while following the zero/low-code principle. Techniques such as cross-validation, class balancing, and artificial augmentation of training data are also considered in the pipeline.

Major comments:

1. Have the authors ever tried brand new data sets on the tasks in Table 1? What's the performance on a brand new data set?
2. As the authors explained, the system which follows zero- and low- code principles is designed for

doctors and other experts with not much computer science background, how to understand and try to tune the parameters in 'yaml' files easily for customized training? Maybe they need a more detailed manual except from the comments in the 'yaml' file.

Reviewer #3 (Remarks to the Author):

The authors address a critical problem in translation of DL algorithms into clinical practice by centralizing the available tools into one platform that can be used by expert and non-expert medical researchers to analyze radiologic and pathologic digitized images. The authors classify their approach to translation of ML to users of various levels of expertise with data science and computer science as tools that can be classified as applications, libraries, toolkits, and frameworks. The authors previously published methods for image viewing and annotation and have expertise in democratizing image analysis tools. In current manuscript, they provide a 'framework' for deep learning that will allow clinical researchers with almost a safeguard to perform quality research in applying DL to clinical problems on a wide range of platforms that include medical images to very large size of data that comes in pathologic images. This has the potential to significantly improve the quality of many studies published in the field. The built in corrections with the author's new GaNDFL framework include performing nested cross validation, addressing class imbalance, and artificial augmentation of training data. Making these parameters as part of a cohesive package will significantly improve the reproducibility of published algorithms from clinician researchers, which per multiple systematic reviews demonstrates significant systematic deficiencies in being adherent to established standards such as CLAIM and TRIPOD and demonstrate significant bias. The authors also propose that GaNDFL will be applicable to multiple DL tasks that are used in medical image analysis, which include segmentation, regression, and classification, therefore this method will subject a large field research in medical imaging to adhere to high level of standards just because they are part of the GaNDFL package. Overall, this paper will significantly affect the research in the field of clinical translation of DL from research labs into actual use of DL in practice. This method provides democratization of DL tools to clinical researchers, standardization of DL tool uses with focus on implementing high standards for appropriate use, and also provides a variety of different tools to diversify approaches toward problems in medical image analysis.

Introduction:

Page 2 second paragraph: The authors state that multiple software packages can be confusing to the less experienced user. While I agree that confusion is not welcome, but any seasoned researcher will move past confusion and will figure out how to use the package that is critically needed in their clinical environment. I recommend to focus on the real problem of having numerous software packages – when the tools are not centralized and standardized, they can first be not detected and thus not used. Lack of centralization of tools makes it difficult to test them and their applicability to datasets outside of the ones they were developed for. This results in wasting of significant effort by different research groups in development of algorithms that will never be applied into clinical practice.

While GaNDFL provides assistance to clinical researchers in many aspects of appropriate application of DL to clinical problems, does this package address the problem of missing data, which is common in real clinical scenarios. It would be great for the authors to discuss how missing data being addressed in their framework.

The authors discuss that GaNDFL allows end-to-end processing of medical images which includes pre- and post-processing steps in a cohesive and reproducible manner. How is it different from well accepted package PyRadiomics?

Results:

In the segmentation section, the authors demonstrate the versatility of GaNDLF across different medical images with excellent segmentation results for brain extraction, glioblastoma segmentation, brain anatomical region segmentation, segmentation of breast tissue into different compartments, healthy and abnormal lung tissue (which included both detection of lung nodules/ie cancer screening and also detecting abnormal lung parenchyma in COVID-19), identification of fundus on retinal scans, dental quadrants on maxillofacial imaging, and abnormal regions on histology slides. This wide variability of imaging modalities and applications of different deep learning algorithms proves the robustness of the GaNDLF platform. It would be great if the authors discuss how they ensured quality of algorithm performance by addressing nested cross validation, addressing class imbalance, artificial augmentation of training data, missing data, appropriate data splitting for training testing and validation in these scenarios. How were the different training strategies such as loss function and optimizers were selected in these different approaches.

Also, was the analysis performed by experienced data scientist or by a clinician researcher. Do the authors think there will be differences in results if they were performed by a clinician researcher as compared to data scientist? Is this a topic that should be addressed in this manuscript?

In the regression section, prediction of brain age is a very appropriate task. Similar detail would be helpful as for segmentation section, how did GaNDLF platform ensure quality of results and would the results be different if the analysis was performed by the experienced data scientist versus less experienced clinician researcher?

Classification section utilized complex tasks such as prediction of eGFRvIII mutation status in GBM, presence of diabetic foot ulceration, and infiltration of TILs into tumor from histology slides. These are complex tasks and demonstrate reasonable predictive values in line with published results.

For each of the algorithms, it is great to see the results and the details in the supplementary material are very helpful. Not all of the descriptions contain all the elements of TRIPOD or CLAIM criteria. I would recommend to include details of algorithm performance and training based on TRIPOD criteria for each challenge. This information will be helpful for researchers that try to reproduce these results although it is not really essential to make the main point of the manuscript. The authors make their point clearly but for the sake of completeness, these details will be very welcome by the community.

One of the questions that would be great to address in the discussion is how do the authors suggest that experienced DL researchers can keep creativity and originality of approaches when using GaNDLF platform as opposed to coding their own algorithms. Does GaNDLF platform have potential for generating out of the box type solutions without creativity or is it mostly a method for easier DL applications with built in safeguards for good use of DL and making DL papers more reproducible?

RESPONSE TO REVIEWER COMMENTS

We would like to take the opportunity of this document to thank the editor and the reviewers for their positive and thoughtful comments and feedback on our manuscript. In the revised version, we have duly incorporated the feedback provided by them and addressed all their comments to further improve the quality of our manuscript.

We further want to thank the editorial team for appreciating the quality of our manuscript and offering us the opportunity to submit the revised version of our manuscript.

*Below we provide a point-to-point response to each of the reviewer comments, including verbatim their original remarks. The reviewers' comments are reproduced **in blue font** followed by our responses in black font. Note that all changes in the manuscript text are denoted by the following color schema, both in the manuscript and in the response document:*

Black text current unchanged text

Green text new or added text

~~Red strikethrough~~ text deleted text

REVIEWER # 1

This paper and its corresponding software framework mainly aim to develop an 'All-in-One' Pytorch-based python package that contains everything needed to perform a stream-lined deep learning training with - Data loader; Built-in module for data preprocessing and data augmentation for common medical imaging classification and segmentation tasks; a collection of well-established (however lacking some of latest ones) neural network models, e.g. UNet, ResNet, etc; a universal predefined training workflow that can be controlled by simply edit the configuration file; logging mechanism to monitor the training process; included some capability for explainable AI, e.g. Grad-Cam

It is mainly designed for easy use of DL to train and test DL models in a clinical environment with low-programming requirements. This is however very similar to other frameworks, e.g. Monai. Essentially, this framework collected the code from other open source projects and put it all together and linked them with an abstracted API layer.

RESPONSE:

We thank the reviewer for the insights into our work and appreciating its value. We really appreciate the depth of the collective comments provided below, and we were happy to see that the reviewer has identified the same points we were working on to further strengthen GaNDLF since the initial manuscript submission. We truly believe that by addressing these comments our manuscript has now been further improved.

In terms of the specific point, we agree that GaNDLF brings together functionality/code from multiple other open-source projects using an abstracted API, as we did not want to replicate work

of others but instead leverage existing software toolkits (including MONAI) and built on top of them while acknowledging them appropriately. Our intention is to provide this exact abstraction layer to provide the value of the streamlined DL training to the community at large, as current software development efforts do not offer such functionality, but rather software toolkits that require computational expertise to make use of them. Along these lines, for both experienced and novice researchers, GaNDF provides well-validated pipelines leveraging the power of DL for generating baseline results in a quick and reproducible manner.

To address this point, we have now change the last paragraph of the “Introduction” section to read as follows:

“Here, we introduce the **GenerAlly Nuanced Deep Learning Framework (GaNDF)** as a community-driven open-source framework by MLCommons, which is an industry-academic partnership aiming to accelerate the adoption of machine learning innovation to benefit the larger community, to enable both **clinical** and **computational researchers** address various AI workloads (such as segmentation, regression, and classification), while producing robust DLAI models without requiring extensive computational experience. This is done by focusing on ensuring that AI algorithms and pipelines follow paradigms adhering to best practices established by the greater ML community, and leveraging existing collaborative efforts in the space (such as the MLCommons’ MedPerf⁶⁴ - medperf.org). Such practices include: (i) nested cross-validation^{59,60}; (ii) handling class imbalance⁶¹; and (iii) artificial augmentation of training data^{62,63}. Additionally, GaNDF incorporates capabilities to handle **end-to-end** processing (i.e., pre- and post-processing steps) in a cohesive and reproducible manner to contribute towards democratizing AI in healthcare, while these best ML practices are at the forefront during training and inference. We have developed GaNDF has been developed in PyTorch/Python as an abstraction layer that incorporates widely used open-source libraries and toolkits (such as MONAI⁴³) that can help researchers generate robust AI models quickly and reliably, facilitating reproducibility⁶⁵ and being consistent with the criteria of findability, accessibility, interoperability, and reusability (FAIR)⁶⁶. Furthermore, the flexibility of its codebase permits GaNDF to be used across modalities (e.g., 2D/3D radiology scans, and 2D multi-level histology whole slide images (WSI)), and has scope and functionality for integrating other clinical data (such as genomics and electronic health records) in the future, thus taking current ~~added~~ clinical diagnostics to the next frontier of quantitative integration.”

Apart from the above, please find below detailed comments:

1. Currently, the models contain a limited number of models, which are not up to date. There are no transformer-based models included in the framework. Although the framework is designed to be easy to extend, however, this highly relies on the maintainers regarding the quality of the implementation.

RESPONSE:

We thank the reviewer for this comment. To address this point, we would like to first differentiate between DL architectures and DL models, by defining a DL architecture the network topology that (once trained) results in a trained DL model.

It is true that GaNDF, as a community-driven effort is being as strong as its community contributions, which have been continuously growing. Since the submission of this manuscript,

GaNDLF has been absorbed by MLCommons [<https://github.com/mlcommons/GaNDLF>], which is an industry-academic partnership aiming to accelerate the adoption of machine learning innovation to benefit the larger community. Members of MLCommons include Google, Meta, Intel, NVidia, RedHat, IHU Strasbourg, Dana Farber Cancer Institute, University of Pennsylvania (among others), that have appropriate expertise to build relevant community contributions as deemed appropriate. A complete list of MLCommons members can be found at <https://mlcommons.org/>. Through this strategic absorption, GaNDLF aims at adopting best practices (i.e., MLCube) developed by the industry, while having its codebase maintained by the MLCommons' full-time engineers, ensuring quality and sustainability. Thus far, GaNDLF has contributions from multiple organizations (academic institutions, and industrial entities), all of whom are included as coauthors of the manuscript.

Specifically about newer transformer-based architectures, since the submission of the manuscript, GaNDLF has received such relevant contributions* focusing on the addition of network architectures, and others that grow continuously (as explicitly indicated in the "major feature additions" shown at the bottom of this document). Furthermore, since its absorption by MLCommons, GaNDLF is now offering functionality related to 1) direct containerization using the Docker, Singularity, and MLCube standards and 2) direct connections to MedPerf towards facilitating the orchestration of algorithmic validation in an effective, coordinated, federated manner. Further ensuring quality contributions, as the maintainers of the framework we seek the automation of quality assurance through several well-known robust software development practices, such as unit testing (to tests of individual functional components of the software, to ensure that implementation changes do not break the usage contract established by that component), system testing (larger-scale tests of software functionality, to test the usage of the software in a broader way that more closely correlates to real usage), and reporting code coverage (a metric collected during testing, reflecting how much of the codebase is traversed). These are now added a new subsection under "Methods" as "Software Development Practices":

"GaNDLF incorporates several well-known robust software development measures¹⁶⁸ to ensure ongoing software quality in the presence of community contributions. These measures include the following:

- **Unit testing** refers to tests of individual functional components of the software, to ensure that implementation

changes do not break the usage contract established by that component. These units are the smallest relevant units of functionality, and testing these helps ensure that bugfixes, feature additions, and performance optimizations do not cause breaking changes to basic calculations made by the software, such as those that would impact model training. GaNDLF includes extensive unit tests for all custom functionality which is built atop other libraries.

- **System testing** refers to larger-scale tests of software functionality, to test the usage of the software in a broader way that more closely correlates to real usage. GaNDLF's test suite includes extensive system tests, including tests of each major usage mode (training, inference, data preparation, and so on), and tests for each model architecture across types of data (such as radiology and histology images) and types of workloads (such as classification, regression, and segmentation) as appropriate. GaNDLF's test suite requires all tests to pass before code can be committed to the repository, and changes cannot be committed to the code repository if any tests fail for any reason.

- **Automated test coverage reports** are metrics collected during testing, reflecting how much of the codebase is traversed. High code coverage indicates that more individual

components, functions, and conditional branches of the software have been tested. GaNDFL automatically reports code coverage changes on any incoming contribution and flags changes that decrease code coverage for further review. While the above tests cover code-level reliability, it is difficult to infer reliability regarding performance of the models produced by GaNDFL, in part due to stochasticity of the training process. We are actively working on additions to the automated test suite that would measure performance of each model on small sample datasets, and flag contributions that cause drops in performance for further review.

• **Continuous deployment** via containerization using the Docker, Singularity, and MLCube standards.”

* github.com/mlcommons/GaNDFL/blob/master/GANDLF/models/transunet.py
github.com/mlcommons/GaNDFL/blob/master/GANDLF/models/unetr.py

2. Zero- and Low- code solution is targeting a small group of clinical audience, and provides a ‘click-to-run’ way to try out some deep learning models. However, there is a lack of a mechanism for optimizing the hyper-parameters which are the keys to performance.

RESPONSE:

We thank the reviewer for this comment and we would like to highlight that we now have a mechanism to optimize and tune hyper-parameters to maximize algorithmic performance per use case. Information on how to make best use of this mechanism is now presented in the software documentation: <https://mlcommons.github.io/GaNDFL/usage#customize-the-training>. This has now been added in the manuscript (“Methods” section, “Modularity & Extendibility” subsection):

“A description of GaNDFL’s software stack, modularity, and extendibility is hereby provided, as well as how the lower-level libraries are utilized to create an abstract user interface, which can be customized based on the application at hand. Following this, the flexibility of the framework from a technical point-of-view is chronicled, which illustrates the ease with which new functionality can be added, *and further details on customizing the entire processing pipeline can be found in the software documentation at mlcommons.github.io/GaNDFL.*”

3. Not support multiple modality inputs. Currently, it seems only supports single inputs (with multichannel support).

RESPONSE:

Indeed, the results presented in the submitted manuscript are from studies of “single inputs” (i.e., either radiology or histology images) with multi-channel support (for example, multiple MRI sequences considered in tandem). We have now explicitly mentioned this as one of the current limitations of the framework (in the discussion sections).

However, we do not consider the support of multiple modality inputs being subject to the framework, but mostly subject to the design of a study and the fusion approach used to combine multiple modalities. For example, one could design a study using GaNDFL considering ‘early’ or ‘late’ fusion. By ‘early’ fusion we refer to the creation of a joint representation of radiology and histology images at the input level (i.e., before feeding to the GaNDFL training mechanism), e.g.,

by vector concatenation following feature extraction, or bilinear pooling. Furthermore, one could also design a study using 'late' fusion by training an independent model (even of different DL architecture) via GaNDFL for each of the considered modalities (radiology and histology) and then aggregate the model predictions at the decision level, for a final prediction. This aggregation can happen by averaging, majority voting, Bayes-based rules [*], or learned models, e.g., a multilayer perceptron [**].

The new related additions in the manuscript's "Discussion" section's 6th paragraph:

"Although GaNDFL has been evaluated across imaging modalities (on radiology and histology images) using single inputs (i.e., either a single radiology or histology image) or with multi-channel support (i.e., multiple MRI sequences considered in-tandem), so far, its application has been limited to workloads related to segmentation, regression, and classification, but not towards synthesis, semi/self-supervised training¹⁰⁵ or physics-informed modeling. Expanding the application areas would further bolster the applicability of the framework. Additionally, application to datasets representing analysis of 4D images (such as dynamic sequences or multi-spectral imaging) has not yet been evaluated. Also, a mechanism to enable aggregation of various models (i.e., train/infer models of different architectures concurrently) is not present, which have generally shown to produce better results^{7-10,45,106,107}. Mechanisms that enable AutoML¹⁰⁸⁻¹¹¹ and other network architecture search (NAS) techniques¹¹² are tremendously powerful tools that create robust models, but are currently not supported in GaNDFL. Finally, application of GaNDFL to other data types, such as genomics or electronic health records (EHR), which would allow GaNDFL to further inform and aid clinical decision making by training multi-modal models, has not been fully explored yet but it is considered as current work in progress."

* Ramanathan, T.T., M. Hossen, and M. Sayeed, Naïve Bayes Based Multiple Parallel Fuzzy Reasoning Method For Medical Diagnosis. *Journal of Engineering Science and Technology.*, 2022. 17(1): p. 0472-0490.

** Bertsimas, D. and H. Wiberg, *Machine learning in oncology: methods, applications, and challenges.* *JCO Clinical Cancer Informatics*, 2020. 4.

4. Only support a very standard supervised training pipeline. Many update-to-date methods, inc. GAN, self-supervised, etc. are not supported.

RESPONSE:

This is a well-taken point, and indeed something we are currently actively working on. So far we have focused on supervised training workloads relating to segmentation, classification, and regression, to highlight the generalizability of the framework across different workloads/tasks, while we also highlight the current limitations of the framework's functionality. To further highlight the limitations put forth from this point, we have now made the following change in the manuscript's "Discussion" section's 6th paragraph:

"Although GaNDFL has been evaluated across segmentation, regression, and classification workloads (across imaging modalities, i.e., on radiology and histology images), its functionality is expected to be further extended towards synthesis (i.e., GANs, diffusion models), semi/self-supervised training¹⁰⁵, or physics-informed modeling."

5. Similarly, it does not support iterative DL, etc.

RESPONSE:

We are not entirely sure what the reviewer refers to by “iterative DL”, as we could not find a specific citation that described this term. However, in our attempt to address this comment, based on intuition we understand this point as referring to one of the following 3 options. Notably, all these options are supported in GaNDFL with associated documentation:

- i. Transfer learning, i.e., using weights from a different training process to make current training work quicker and better. GaNDFL supports this, for models trained on both 2D and 3D data.*
- ii. Resuming training from a specific checkpoint, i.e., using model checkpoint from a specific epoch that has better scores to fine-tune model performance of subsequent training processes.*
- iii. Human-in-the-loop, i.e., where a human operator makes changes in the data, specifically the ground truth to improve the predictions during training.*

6. It is not clear how the framework handles large datasets, particularly imaging data in 3D, in an efficient way, given what they mentioned the main usage of this framework might be in resource-constrained environments.

RESPONSE:

With regard to this point, we would like to clarify that we consider it as 2 distinct areas of focus a) handling large data during training, b) handling large data during inference (which is associated to GaNDFL’s ability to run models in resource-constrained environments).

In terms of (a), we would like to highlight the main aspects of GaNDFL’s data flow pipeline that allows efficient processing of large datasets (such as large histology images or large 3D volumes) that relate to i) patch-based training, which allows the model to operate on smaller “chunks” of the data at a single instance (the size and overlap of these chunks can be customized by the user), and ii) lazy data loading, which allows GaNDFL to only read the datasets into the memory during computation, and immediately deallocate the memory once it is used. We have further added appropriate new text in the manuscript (in the “Data Flow Diagrams” subsection under the supplementary materials), that reads as:

“The data flow diagram of GaNDFL leverages 2 main ideas that allows efficient processing of large datasets (such as histology images or large 3D volumes): i) patch-based training and inference, which allows the model to operate on smaller “chunks” of the data at a single instance, which allows the model to operate on the full gamut of images – the size and overlap of these chunks can be customized by the user, ii) lazy loading of the datasets themselves, which allows GaNDFL to only read the datasets into the memory during computation, and immediately deallocate the memory once it is used.”

In terms of (b), since the submission of this manuscript, GaNDFL has already been referenced in a MICCAI proceedings publication explicitly focusing on automatically performing graph-level optimizations of trained models through a tight integration with OpenVINO. We have also now

added a reference of this work (reference 167) under the “Model Optimization” subsection in “Methods”.

7. There is a lack of distributed and parallel training support, which limits the usage in real environments with relatively large datasets, 3D models and large Transformer-based models.

RESPONSE:

This is another well taken point, and we would like to highlight a recent contribution to GaNDF (done on the 21st of September, i.e., prior to receiving these reviewer comments) that allows parallel training across multiple GPUs in a single system: <https://github.com/mlcommons/GaNDF/pull/503>. We have now also explicitly added this in the manuscript (“Methods” section, “Training Mechanism” subsection, “Zero-code Principle”), that reads as:

“The subject identifiers are used to randomly split the entire dataset into training, validation, and testing subsets, using nested k-fold cross-validation¹⁴⁶. The training can be configured to run on multiple DL accelerator cards, such as GPU or Gaudi.”

However, distributed training across multiple devices in a high-performance computing (HPC) environment is a different problem to address, due to the numerous ways one can setup and configure a HPC cluster (including the complexity of various non-standardized job schedulers, such as SLURM, OpenHPC, Kubernetes, SGE). Nevertheless, in order to account for this discrepancy, we have ensured that GaNDF allows multiple training jobs to be submitted in a straightforward manner using the command line interface. Specific instructions can be found in the associated documentation: <https://mlcommons.github.io/GaNDF/usage#parallelize-the-training>.

REVIEWER # 2

This manuscript proposes an open-source framework named Generally Nuanced Deep Learning Framework (GaNDLF) for end-to-end deep learning (DL) based medical image analysis applications. It's a general framework which could process both radiology and histology data for a variety of DL workloads/tasks while following the zero/low-code principle. Techniques such as cross-validation, class balancing, and artificial augmentation of training data are also considered in the pipeline.

RESPONSE:

We thank the reviewer for their positive comments and the insight that they have provided.

Major comments:

1. Have the authors ever tried brand new data sets on the tasks in Table 1? What's the performance on a brand new data set?

RESPONSE:

We thank the reviewer for their question, and we have now clarified in the manuscript that the results shown in Table 1 are obtained from hold out data (i.e., brand new data – not included in the training dataset). We have now even further added an indication for when out-of-sample data (i.e., from sources/sites that have not contributed any data for the model training) have been used for the algorithmic evaluation, along with the appropriate text changes in the “Results” section, reading as:

“The reported results for all the performed experiments are on the unseen testing (or holdout⁶⁷) cohorts for each application, and collectively shown in Table 1.”

2. As the authors explained, the system which follows zero- and low- code principles is designed for doctors and other experts with not much computer science background, how to understand and try to tune the parameters in ‘yaml’ files easily for customized training? Maybe they need a more detailed manual except from the comments in the ‘yaml’ file.

RESPONSE:

We thank the reviewer for this well taken point. We would like to mention that such a detailed “user manual” providing an understanding of the tuned parameters is now provided in the associated documentation: <https://mlcommons.github.io/GaNDLF/usage#customize-the-training>. We have now also noted this explicitly in our manuscript under the “Methods” section, “Modularity and Extendibility” subsection, as:

“A description of GaNDLF's software stack, modularity, and extendibility is hereby provided, as well as how the lower-level libraries are utilized to create an abstract user interface, which can be customized based on the application at hand. Following this, the flexibility of the framework from a technical point-of-view is chronicled, which illustrates the ease with which new functionality can be added, and further details on customizing the entire processing pipeline can be found in the software documentation at mlcommons.github.io/GaNDLF.”

REVIEWER # 3

The authors address a critical problem in translation of DL algorithms into clinical practice by centralizing the available tools into one platform that can be used by expert and non-expert medical researchers to analyze radiologic and pathologic digitized images. The authors classify their approach to translation of ML to users of various levels of expertise with data science and computer science as tools that can be classified as applications, libraries, toolkits, and frameworks. The authors previously published methods for image viewing and annotation and have expertise in democratizing image analysis tools. In current manuscript, they provide a 'framework' for deep learning that will allow clinical researchers with almost a safeguard to perform quality research in applying DL to clinical problems on a wide range of platforms that include medical images to very large size of data that comes in pathologic images. This has the potential to significantly improve the quality of many studies published in the field. The built in corrections with the author's new GaNDF framework include performing nested cross validation, addressing class imbalance, and artificial augmentation of training data. Making these parameters as part of a cohesive package will significantly improve the reproducibility of published algorithms from clinician researchers, which per multiple systematic reviews demonstrates significant systematic deficiencies in being adherent to established standards such as CLAIM and TRIPOD and demonstrate significant bias. The authors also propose that GaNDF will be applicable to multiple DL tasks that are used in medical image analysis, which include segmentation, regression, and classification, therefore this method will subject a large field research in medical imaging to adhere to high level of standards just because they are part of the GaNDF package. Overall, this paper will significantly affect the research in the field of clinical translation of DL from research labs into actual use of DL in practice. This method provides democratization of DL tools to clinical researchers, standardization of DL tool uses with focus on implementing high standards for appropriate use, and also provides a variety of different tools to diversify approaches toward problems in medical image analysis.

RESPONSE:

We thank the reviewer for their positive comments, especially about considering our framework to "significantly improve the quality of many studies published in the field" and "significantly improve the reproducibility of published algorithms from clinician researchers".

Introduction:

1. Page 2 second paragraph: The authors state that multiple software packages can be confusing to the less experienced user. While I agree that confusion is not welcome, but any seasoned researcher will move past confusion and will figure out how to use the package that is critically needed in their clinical environment. I recommend to focus on the real problem of having numerous software packages – when the tools are not centralized and standardized, they can first be not detected and thus not used. Lack of centralization of tools makes it difficult to test them and their applicability to datasets outside of the ones they were developed for. This results in wasting of significant effort by different research groups in development of algorithms that will never be applied into clinical practice.

RESPONSE:

We thank the reviewer for their keen insight regarding this problem. Our motivation for developing GaNDF revolves around creating algorithms that can be executed by people outside the direct GaNDF software contributors, towards facilitating easier clinical translation. We are hoping to grow a community around creating clinically relevant workflows for DL tasks by leveraging

GaNDLF. Therefore, by creating tools standardized within the same infrastructure (GaNDLF) for the entire community to leverage, we anticipate the cost and time of creating algorithms to be substantially reduced and hence put efforts in meaningfully translating methods into the clinical practice rather than trying to make a tool to work. We have also added relevant information in the manuscript's "Discussion" section (last paragraph), that reads as:

"Finally, by creating tools standardized within the same infrastructure (GaNDLF) for the entire community to leverage, we anticipate the cost and time of creating algorithms to be substantially reduced and hence put efforts in meaningfully translating methods into the clinical practice rather than trying to make a tool to work."

2. While GaNDLF provides assistance to clinical researchers in many aspects of appropriate application of DL to clinical problems, does this package address the problem of missing data, which is common in real clinical scenarios. It would be great for the authors to discuss how missing data being addressed in their framework.

RESPONSE:

We thank the reviewer for their keen insight into the problem of missing data. This is indeed an important problem that needs due consideration by the community. There are 2 factors need to be considered for this: i) since computational researchers require distinct input data specifications for algorithmic development, that is what GaNDLF focusses on, and ii) dealing with missing data (either via image synthesis or via data augmentation techniques) is indeed an important and clinically relevant problem, and it is something that an investigator interested in this study could define using GaNDLF. As this is an active research topic in the field of clinical machine learning, we are hoping to collaborate with researchers that are well-versed with this topic to further the goals of leveraging GaNDLF as the "go-to" tool when performing DL for clinical tasks. We would further want to mention to the reviewer that we are currently working on integrating image synthesis via GANs (as explicitly indicated in the "major feature additions" shown at the bottom of this document) that would allow us to address a specific set of missing data, but this is planned to be available in GaNDLF by Q2 2023.

3. The authors discuss that GaNDLF allows end-to-end processing of medical images which includes pre-and post-processing steps in a cohesive and reproducible manner. How is it different from well accepted package PyRadiomics?

RESPONSE:

We thank the reviewer for their question. Indeed, the development of GaNDLF was motivated by the absence of a single package that would allow definition of a complete clinical workflow (starting from data preparation and continuing to pre- and post-processing) in a reproducible manner for deep learning tasks that could be applied across various healthcare imaging domains. In terms of GaNDLF's comparison to PyRadiomics, the latter is a package for image feature extraction that allows users to obtain quantitative measurements from images. It neither allows users to customize the pre- and post- processing steps for the input image(s) to ensure consistency in the extracted measurements, nor allows the model training, workflow generation, evaluation mechanisms, cross-validation techniques. Additionally, the output of PyRadiomics is a collection of radiomic features, which can be integrated into models trained by GaNDLF to provide additional context.

Results:

4. In the segmentation section, the authors demonstrate the versatility of GaNDLF across different medical images with excellent segmentation results for brain extraction, glioblastoma segmentation, brain anatomical region segmentation, segmentation of breast tissue into different compartments, healthy and abnormal lung tissue (which included both detection of lung nodules/ie cancer screening and also detecting abnormal lung parenchyma in COVID-19), identification of fundus on retinal scans, dental quadrants on maxillofacial imaging, and abnormal regions on histology slides. This wide variability of imaging modalities and applications of different deep learning algorithms proves the robustness of the GaNDLF platform. It would be great if the authors discuss how they ensured quality of algorithm performance by addressing nested cross validation, addressing class imbalance, artificial augmentation of training data, missing data, appropriate data splitting for training testing and validation in these scenarios. How were the different training strategies such as loss function and optimizers were selected in these different approaches.

RESPONSE:

We thank the reviewer for their keen insight into the process of training robust DL models. Specific training strategies (loss function, optimizers, augmentation, and so on) are dependent on the application itself, and their configuration used in the showcased experiments have been leveraged using existing literature. Each of the choices on specific hyper-parameters, since we were reproducing previously published work, were based on the set of parameters defined in the original studies. We further provide multiple samples for different workloads for a user to get started [<https://github.com/mlcommons/GaNDLF/blob/master/samples>], which a user can customize further for optimal performance.

We would also like to highlight that we now have a mechanism to optimize and tune hyper-parameters towards maximizing algorithmic performance per use case. Information on how to make best use of this mechanism is now presented in the software documentation: <https://mlcommons.github.io/GaNDLF/usage#customize-the-training>. This has also been explicitly added in the manuscript (“Methods” section, “Modularity & Extendibility” subsection), and reads as:

“Further details on customizing the entire processing pipeline (including hyper-parameter tuning and optimization) can be found in the software documentation at mlcommons.github.io/GaNDLF.”

Regarding class imbalance, we have now added a new sub-section “Handling Class Imbalance” under the “Training Mechanism” section under “Methods” that reads as:

*“**Handling Class Imbalance.** Class imbalance, i.e., where the presence of one class is significantly different in proportion to another, is a common problem in healthcare informatics^{149,150}. To address this issue, GaNDLF allows the user to set a penalty for the loss function¹⁵¹, which is inversely proportional to the classes being trained on. The penalty weights for the loss function will be defined as:*

$$= 1 -$$

Where ‘ p_c ’ is the penalty for class ‘ c ’, and ‘ n_c ’ is the number of instances of the presence of class ‘ c ’ in the total number of samples ‘ N ’.

For example, for a classification workload using 100 cases, if there are 10 from class 0 and 90 from class 1, the weighted loss will get calculated to 0.9 for class 0 and 0.1 for class 1.

This basically means that the misclassification penalty during loss back-propagation for class 0 (i.e., the “rarer” class) will be higher than that of class 1 (i.e., the more “common” class). The analogous process can be done for segmentation workloads as well. We recognize that this approach might not work for all problem types, and thus we have mechanisms for the user to specify a pre-determined loss penalty for greater customization.”

5. Also, was the analysis performed by experienced data scientist or by a clinician researcher. Do the authors think there will be differences in results if they were performed by a clinician researcher as compared to data scientist? Is this a topic that should be addressed in this manuscript?

RESPONSE:

We thank the reviewer for the valuable comment. Different applications were done by clinical researchers and by data scientists, and answering the question regarding which one of the groups would outperform the other in quantitative metrics might not be possible. The clinical researchers bring in the domain expertise (for example, which data augmentations are most likely to occur for a specific application), and data scientists bring in the DL training expertise (i.e., which loss function will be most appropriate for a specific application). In reality, the best results are observed when both these groups of the scientific community work together to further our understanding of healthcare. Towards that end, we hope that GaNDLF can provide a common frame of reference for both these user groups. We would also like to mention that because of the inherent randomness in a deep learning training session, even the same researcher running the same experiment will see slightly different results each time. To address this, we are adding an option for determinism during a training process (as indicated in the “major feature additions” shown at the bottom of this document). We have added relevant text under the “Discussion” section (3rd paragraph) that reads as:

“Furthermore, GaNDLF provides the means to DL researchers/developers to distribute their methods in a reproducible way to the wider community, thereby expanding their application across various problem domains with relative ease, and providing re-usable components (Figure 2) that can be combined to create customized solutions. Ideally, we anticipate the best results when both these groups of the scientific and clinical community bring their expertise together to further our understanding of healthcare. Towards this end, GaNDLF can provide a common frame of reference for both these user groups. By creating tools standardized within the same infrastructure (GaNDLF) for the entire community to leverage, we anticipate the cost and time of creating algorithms to be substantially reduced and hence put efforts in meaningfully translating methods into the clinical practice rather than trying to identify and/or make a tool to work.”

6. In the regression section, prediction of brain age is a very appropriate task. Similar detail would be helpful as for segmentation section, how did GaNDLF platform ensure quality of results and would the results be different if the analysis was performed by the experienced data scientist versus less experienced clinician researcher?

RESPONSE:

We thank the reviewer for appreciating the appropriateness of this application. For all the segmentation applications discussed in the manuscript, there was at least 1 holdout testing set

which was used for performance evaluation. In most cases, the nested training mechanism ensured that there was randomization in that set as well. However, we recognize that the most robust mechanism of performing quality assurance of algorithmic generalizability would be to test the performance of trained models on completely out-of-sample or out-of-distribution cases. We have already shown an example of such evaluation through the integration of GaNDFL with [MedPerf \[https://doi.org/10.48550/arXiv.2110.01406\]](https://doi.org/10.48550/arXiv.2110.01406) during the Federated Tumor Segmentation Challenge [\[https://doi.org/10.48550/arXiv.2105.05874\]](https://doi.org/10.48550/arXiv.2105.05874), where models trained using GaNDFL were evaluated in-the-wild on data of unknown sources. Furthermore, GaNDFL has been absorbed by MLCommons, which is an industry-academic partnership aiming to accelerate the adoption of machine learning innovation to benefit the larger community. Members of MLCommons include Google, Meta, Intel, NVidia, RedHat, IHU Strasbourg, Dana Farber Cancer Institute, University of Pennsylvania (among others), that have appropriate expertise to build relevant community contributions as deemed appropriate. A complete list of MLCommons members can be found at <https://mlcommons.org/>. Through this absorption, GaNDFL aims at adopting best practices (i.e., MLCube) developed by the industry, while having its codebase maintained by the MLCommons' full-time engineers, ensuring quality and sustainability.

As also mentioned in the previous comment, the best results would be observed when both the clinical and computational researchers work together to further our understanding of healthcare. Towards that end, we hope that GaNDFL can provide a common frame of reference for both these user groups and that with the MedPerf integration would allow the evaluation of trained models happening "in the wild" (under actual real-world conditions). Additionally, once we have incorporated determinism in the training process of GaNDFL (see "major feature additions", shown at the bottom of this document), these differences would hopefully be less apparent.

7. Classification section utilized complex tasks such as prediction of eGFRvIII mutation status in GBM, presence of diabetic foot ulceration, and infiltration of TILs into tumor from histology slides. These are complex tasks and demonstrate reasonable predictive values in line with published results.

RESPONSE:

We thank the reviewer for appreciating our work.

8. For each of the algorithms, it is great to see the results and the details in the supplementary material are very helpful. Not all of the descriptions contain all the elements of TRIPOD or CLAIM criteria. I would recommend to include details of algorithm performance and training based on TRIPOD criteria for each challenge. This information will be helpful for researchers that try to reproduce these results although it is not really essential to make the main point of the manuscript. The authors make their point clearly but for the sake of completeness, these details will be very welcome by the community.

RESPONSE:

We thank the reviewer for this invaluable comment, and we agree that this is something that researchers in our field need to always address. However, since this manuscript focus is on the software framework (GaNDFL) and not on a particular study (i.e., we do not report any new results but only recreations of previous publications), we think that reporting such criteria here would be out-of-scope. However, since we really agree with the reviewer about raising awareness and

considering the importance of this topic for our community, we have now added a complete new paragraph in the “Discussion” section (7th paragraph), that reads as:

“To facilitate clinical applicability, reproducibility, and translation, in the domain of healthcare AI, published research is essential to adhere to well-accepted reporting criteria. Some of these criteria are: i) CLAIM (Checklist for Artificial Intelligence in Medical Imaging)¹¹³, which outlines the information that authors of medical-imaging AI articles should provide, ii) STARD-AI, which is the AI-specific version of the Standards for Reporting of Diagnostic Accuracy Study (STARD) checklist¹¹⁴, and aims to address challenges related to the original STARD checklist related to the utilization of AI models, iii) TRIPOD-AI and PROBAST-AI, which are the AI versions of the TRIPOD (Transparent Reporting of a multivariable prediction model of Individual Prognosis Or Diagnosis) statement and the PROBAST (Prediction model Risk Of Bias ASsessment Tool)¹¹⁵, and aim to provide standards both for reporting but also for Risk of Bias assessment, raising awareness of the importance in meta-analyses dealing with AI studies, iv) CONSORT-AI and SPIRIT-AI, which are the AI extensions of the CONSORT (Consolidated Standards of Reporting Trials) and SPIRIT (Standard Protocol Items: Recommendations for Interventional Trials), providing guidance for reporting randomized clinical trials¹¹⁶, v) MI-CLAIM (Minimum Information about Clinical Artificial Intelligence Modelling)¹¹⁷ which focuses on the clinical impact and the technical reproducibility of clinically relevant AI studies, vi) MINIMAR (MINimum Information for Medical AI Reporting)¹¹⁸, which sets the reporting standards for medical AI applications by specifying the minimum information that AI manuscripts should include, and vii) Radiomics Quality Score (RQS)¹¹⁹, which outlines 16 criteria by which to judge the quality of a publication on radiomics¹²⁰.”

9. One of the questions that would be great to address in the discussion is how do the authors suggest that experienced DL researchers can keep creativity and originality of approaches when using GaNDLF platform as opposed to coding their own algorithms. Does GaNDLF platform have potential for generating out of the box type solutions without creativity or is it mostly a method for easier DL applications with built in safeguards for good use of DL and making DL papers more reproducible?

RESPONSE:

We thank the reviewer for their keen interest on GaNDLF. We aim to allow researchers to quickly generate baseline results using multiple “out-of-the-box” solutions presented in GaNDLF to check for “signal” in a new dataset, and at the same time provide enough modularity in the codebase, for computational researchers to plug in their algorithms in an easy manner. Almost all the results shown in this paper were done using the former paradigm, and new network architectures, and new pre- and post-processing methods** were contributed by the community. Since GaNDLF is completely based on bare-metal PyTorch, any algorithm written using PyTorch can be imported into GaNDLF with relative ease. In the case of new algorithms being incorporated, GaNDLF ensures that they are checked for computational cohesiveness against all expected data types (2D RGB, 2D grayscale, 3D radiology, single and multi-channel data) so that when they are used by another researcher, it works as an “out-of-the-box” solution. We believe the following section of the “Discussion” section (3rd paragraph) adequately addresses this point (by building upon the response of comment 5):*

“For DL researchers/developers, GaNDLF provides a mechanism for creating customized solutions, robust evaluation of their methods across a wide array of medical datasets that span across dimensions, channels/modalities, and prediction classes, as well as to conduct

a comparative quantitative performance evaluation of their algorithm against well-established built-in network architectures, including, but not limited to, UNet⁹⁰⁻⁹², UNetR (UNet with transformer encoding)⁹³, VGG^{94,95}, DenseNet⁹⁶, ResNet⁹⁷, and EfficientNet⁹⁸. Furthermore, GaNDLF provides the means to DL researchers/developers to distribute their methods in a reproducible way to the wider community, thereby expanding their application across various problem domains with relative ease, and providing re-usable components (Figure 2) that can be combined to create customized solutions. Ideally, we anticipate the best results when both these groups of the scientific and clinical community bring their expertise together to further our understanding of healthcare. Towards this end, GaNDLF can provide a common frame of reference for both these user groups. By creating tools standardized within the same infrastructure (GaNDLF) for the entire community to leverage, we anticipate the cost and time of creating algorithms to be substantially reduced and hence put efforts in meaningfully translating methods into the clinical practice rather than trying to identify and/or make a tool to work.”

* https://github.com/mlcommons/GaNDLF/blob/master/GANDLF/models/imagenet_unet.py

** <https://github.com/mlcommons/GaNDLF/issues/495>

** <https://github.com/mlcommons/GaNDLF/pull/467>

FOR THE INFORMATION OF ALL THE REVIEWERS

MAJOR FEATURE ADDITIONS SINCE INITIAL MANUSCRIPT SUBMISSION

(Note that these are not a comprehensive ChangeLog, which is shown in GaNDLF's GitHub repository)

- Migrated to MLCommons, and integrated with MedPerf
- Histology training and inference support for consideration of microns, allowing multi-resolution training
- Added network topologies (ImageNet pretrained encoders (this has been accepted in Brain Lesion workshop at MICCAI 2022 but is not yet online) [ref], TransUNet, UNet with deep supervision
- Histology inference mechanism for memory-constrained environments
- Resource utilization monitoring functionality
- Integration with OpenFL allowing any GaNDLF model to be directly trainable in a federated learning setting

UPCOMING MAJOR FEATURE ADDITIONS (planned by Q2 2023)

- Synthesizing images with and without pathologies using GANs
- Histology-specific data augmentation based on stain variations
- New compound loss functions (useful for segmentation)
- During training model optimization (added support for accuracy-aware training via Neural Network Compression Framework for 8-bit models)
- Support for models trained with intrinsic privacy support (i.e., DP)
- Support for non-imaging data types (specifically, electronic health records and genomic profiles)
- Support for multi-modal training (train a single model that combines imaging with non-imaging data)
- Functionality for optional determinism in entire workflow

REVIEWERS' COMMENTS:

Reviewer #1 (Remarks to the Author):

I thank the authors for the revised paper. The completeness of the tools has been improved. The focus on usability in design is interesting and is a novel idea. However, my remaining concern is that GaNDLF is much less comprehensive when compared with other frameworks eg MONAI, which includes more advanced models/architectures and has a more extensive user base. Even GaNFDLF aims for a zero-code principle, its installation and setup are not streamlined for clinical users.

Reviewer #2 (Remarks to the Author):

This manuscript proposes an open-source framework named Generally Nuanced Deep Learning Framework (GaNDLF) for end-to-end deep learning (DL) based medical image analysis applications. It's a general framework which could process both radiology and histology data for a variety of DL workloads/tasks while following the zero/low-code principle. Techniques such as cross-validation, class balancing, and artificial augmentation of training data are also considered in the pipeline.

Reviewer #3 (Remarks to the Author):

Dear Authors, thank you for the thorough response to my revision comments. All my questions were addressed. I would like to congratulate you on this exciting work and it will be a significant advance for clinical translation of ML.

Reviewer #1 (Remarks to the Author):

I thank the authors for the revised paper. The completeness of the tools has been improved. The focus on usability in design is interesting and is a novel idea. However, my remaining concern is that GaNDLF is much less comprehensive when compared with other frameworks eg MONAI, which includes more advanced models/architectures and has a more extensive user base. Even GaNDLF aims for a zero-code principle, its installation and setup are not streamlined for clinical users.

RESPONSE:

We thank the reviewer for their keen perspective, and we understand the confusion. Our concept of developing GaNDLF was not meant to create a duplicate effort and offer another comprehensive toolkit, or library, but built upon existing ones (such as MONAI) and contribute a holistic framework targeting both computational and non-computational end-users. To further clarify the relationship of GaNDLF with other applications, libraries, and toolkits (such as MONAI) to the reviewer we would like to point to figure 2, which (as mentioned in the 2nd paragraph of the introduction) offers a schematic stratification of software efforts into a set of well-defined categories to deepen the community's understanding. GaNDLF represents an abstraction layer on top of existing toolkits and libraries (such as PyTorch, Numpy, MONAI) to leverage their functionality in a mechanism that can be defined as a clinical workflow. To ensure clarity about this relationship we have now edited the concluding paragraph of the "Discussion" section to read as:

Due to its flexible software architecture, it is possible to either leverage certain parts of GaNDLF in other applications/toolkits, or leverage functions other toolkits (e.g., MONAI) and libraries to incorporate them within the holistic functionality of GaNDLF.

Reviewer #2 (Remarks to the Author):

This manuscript proposes an open-source framework named Generally Nuanced Deep Learning Framework (GaNDLF) for end-to-end deep learning (DL) based medical image analysis applications. It's a general framework which could process both radiology and histology data for a variety of DL workloads/tasks while following the zero/low-code principle. Techniques such as cross-validation, class balancing, and artificial augmentation of training data are also considered in the pipeline.

RESPONSE:

We thank the reviewer for the positive comments.

Reviewer #3 (Remarks to the Author):

Dear Authors, thank you for the thorough response to my revision comments. All my questions were addressed. I would like to congratulate you on this exciting work and it will be a significant advance for clinical translation of ML.

RESPONSE:

We thank the reviewer for the positive comments and for appreciating the potential of the proposed framework for clinical end-users.